# XRE transcription factors conserved in *Caulobacter* and φCbK modulate adhesin development and phage production

**Maeve McLaughlin**, **Aretha Fiebig***, **Sean Crosson***

Department of Microbiology and Molecular Genetics, Michigan State University, East Lansing, Michigan, United States of America

* fiebigar@msu.edu (AF); crosson4@msu.edu (SC)

**Data Availability Statement:** Raw sequencing data are available in the NCBI GEO database under series accession GSE241057.

## Abstract

The xenobiotic response element (XRE) family of transcription factors (TFs), which are commonly encoded by bacteria and bacteriophage, regulate diverse features of bacterial cell physiology and impact phage infection dynamics. Through a pangenome analysis of *Caulobacter* species isolated from soil and aquatic ecosystems, we uncovered an apparent radiation of a paralogous XRE TF gene cluster, several of which have established functions in the regulation of holdfast adhesin development and biofilm formation in *C. crescentus*. We further discovered related XRE TFs throughout the class *Alphaproteobacteria* and its phages, including the φCbK Caulophage, suggesting that members of this cluster impact host-phage interactions. Here we show that a closely related group of XRE transcription factors encoded by both *C. crescentus* and φCbK can physically interact and function to control the transcription of a common gene set, influencing processes including holdfast development and the production of φCbK virions. The φCbK-encoded XRE paralog, *tgrL*, is highly expressed at the earliest stages of infection and can directly inhibit transcription of host genes including *hfiA*, a potent holdfast inhibitor, and *gafYZ*, an activator of prophage-like gene transfer agents (GTAs). XRE proteins encoded from the *C. crescentus* chromosome also directly repress *gafYZ* transcription, revealing a functionally redundant set of host regulators that may protect against spurious production of GTA particles and inadvertent cell lysis. Deleting the *C. crescentus* XRE transcription factors reduced φCbK burst size, while overexpressing these host genes or φCbK *tgrL* rescued this burst defect. We conclude that this XRE TF gene cluster, shared by *C. crescentus* and φCbK, plays an important role in adhesion regulation under phage-free conditions, and influences host-phage dynamics during infection.

## Author summary

During infection, bacteria and their viruses (i.e. phage) modulate each other's transcription to promote their own fitness. A broadly conserved group of proteins that are commonly engaged in this battle between host and virus are the xenobiotic response element (XRE) family of transcription factors (TFs). We identified a conserved cluster of XRE TF

**Funding:** This work was supported by the National Institute of General Medical Science of the National Institutes of Health under award number R35 GM131762 to S.C. and F32 GM141017 to M.M. The funders had no role in study design, data collection and analysis, decision to publish, or preparation of the manuscript.

**Competing interests:** The authors have declared that no competing interests exist.

genes in *Alphaproteobacteria* and their bacteriophage. In *Caulobacter crescentus* and its phage, φCbK, these closely related transcription factors regulate a common gene set that impacts *Caulobacter* adhesion and phage virion production. We measured transcription of the φCbK genome across an infection cycle and discovered that the phage XRE TF gene, *tgrL*, is highly expressed at the earliest stages of infection, and we present evidence that TgrL enhances φCbK fitness. Our results offer an example of how an evolutionarily related set of transcription factors, found in both a host and its virus, influence host defense mechanisms and viral fitness.

## Introduction

Surface attachment is a highly regulated process that provides fitness advantages for bacteria in a variety of environmental contexts [1–3]. Our recent studies of surface adherence in the freshwater isolate, *Caulobacter crescentus*, have revealed an elaborate network of transcription factors (TFs) that control development of the cell envelope-anchored adhesin known as the holdfast (Fig 1) [4–7]. Among the direct regulators of holdfast development are proteins classified as xenobiotic response element-family (XRE) TFs, which are widely distributed in bacteria, archaea, bacteriophage, and plasmids [8]. XRE proteins stand out as one of the most abundant classes of TFs in the bacterial kingdom, with thousands of entries in the InterPro database [9]. These proteins are known to regulate diverse metabolic processes and environmental responses [10–12], including biofilm development [13,14], oxidative stress resistance [15], phase variation and pigment production [16]. However, the most well-studied members of this TF family are likely the Cro and CI repressors of bacteriophage λ and 434 [17]. These temperate phages can switch between a lysogenic state, in which the phage integrates into the host chromosome as a prophage, and a lytic state, in which the phage replicate and lyse their hosts. Cro and CI regulate the bistable genetic switch that controls the lysogenic-lytic decision circuit [18–21].

To develop a more complete understanding of *C. crescentus* XRE-family holdfast regulators and to investigate their relationship with other XRE-family TFs in the genus, we constructed a *Caulobacter* pangenome (Fig 2), which revealed hundreds of XRE proteins that grouped into

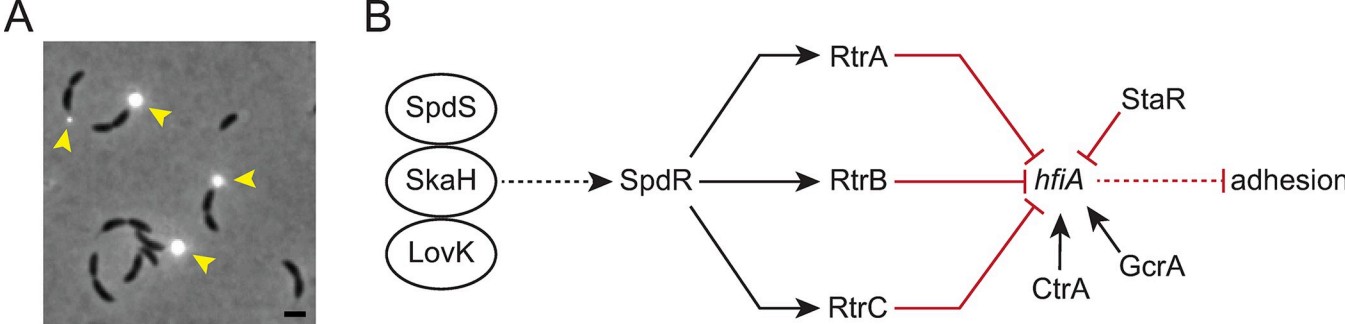

**Fig 1. A network of two-component regulators and XRE-family transcription factors control *Caulobacter* holdfast development. A)** Overlay of phase contrast and fluorescence micrographs of *C. crescentus* CB15 cultivated on peptone-yeast extract (PYE) medium, provides an example of unipolar polysaccharide adhesins in *Alphaproteobacteria*. The *C. crescentus* adhesin, known as the holdfast, is stained with wheat germ agglutinin conjugated to Alexa594 dye (yellow arrowheads). The holdfast is present at the tip of the stalk in a fraction of pre-divisional cells when cultivated in PYE. Scale bar = 1 μm. **B)** Schematic of a regulatory network that regulates transcription of the holdfast inhibitor, *hfiA*. Network includes the essential cell cycle regulators CtrA and GcrA [6,7,22]. A set of interacting sensor histidine kinases (LovK-SkaH-SpdS) activate the DNA-binding response regulator, SpdR. Dashed lines indicate post-transcriptional regulation and solid lines indicate transcriptional regulation. Black arrows indicate activation and red bar-ended lines indicate repression.

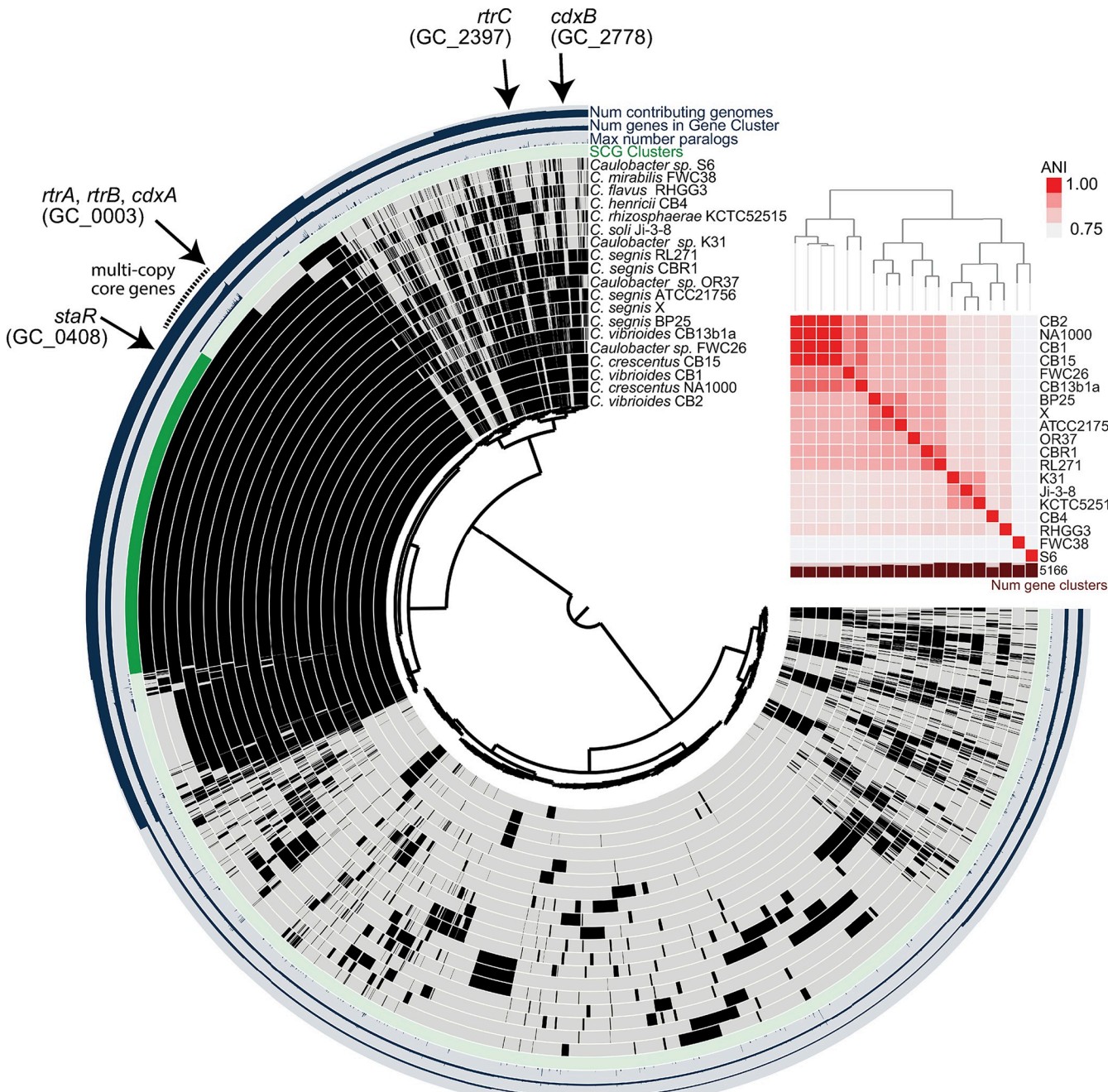

**Fig 2. Conservation of XRE-family adhesion regulators in *Caulobacter*. A)** Pangenome constructed from whole-genome sequences of 19 *Caulobacter* species isolated from freshwater or soil. Internal dendrogram based on shared presence and absence of gene clusters generated using the Markov Cluster (MCL) algorithm (mcl-inflation = 3; min-occurrence = 2); gene clusters organized based on Euclidian distance metric and Ward linkage. Black bars in the internal 19 circles show presence of gene clusters; gray bars indicate absence of gene clusters. Number of genomes that contribute to a gene cluster (from 2–19), number of genes in a gene cluster (from 2–82), and maximum number of paralogs in any single species (from 1–7) are plotted on the outer 3 circles. Genomes are organized by average nucleotide identity (ANI). Dendrogram based on ANI shown above the red heatmap. Single-copy core genes (SCG), i.e. genes present in one copy in all species, are marked in dark green. Multi-copy core genes, i.e. genes present in all species and in more than one copy in at least one species, are marked with a dotted line outside the circle. Gene clusters containing previously reported holdfast regulators *rtrA*, *rtrB*, *rtrC*, and *staR*, and newly-discovered XRE-family transcriptional regulators, *cdxA* (*CCNA_00049*) and *cdxB* (*CCNA_02755*) are marked with black arrows.

dozens of distinct gene clusters. The largest cluster of XRE-family TFs–and the third largest gene family overall–includes known *C. crescentus* holdfast regulators RtrA and RtrB [7]. Some species encode as many as seven paralogs in this gene cluster, which suggests it has undergone a process of radiation in the genus *Caulobacter*. We expanded our genome analysis beyond *Caulobacter* and identified members of this large XRE family across *Alphaproteobacteria* and in a diverse array of bacteriophages that infect *Alphaproteobacteria* including viruses of *Agrobacterium*, *Sinorhizobium*, *Brevundimonas*, and *Caulobacter*.

The proliferation of XRE gene clusters within *Caulobacter*, coupled with their presence across varied *Alphaproteobacteria* phage, prompted us to explore the function of both host and phage XRE genes in the infection cycle of φCbK, a virulent *Caulobacter* phage. Our data reveal that a collection of *C. crescentus* XRE proteins, along with the closely related XRE TF φCbK *gp216* –named *tgrL*–have related DNA binding profiles. These TFs regulate a common set of genes, including a post-translational inhibitor of holdfast development (*hfiA*) [22] and a prophage-like gene transfer agent (GTA) cluster, which is recognized for its ability to package genomic DNA and initiate cell lysis [23]. Deletion of selected host XRE TFs reduced the burst size of φCbK, and this burst size defect could be rescued by expressing φCbK *tgrL* from the chromosome. A time series transcriptomic analysis showed that φCbK *tgrL* is highly expressed at the earliest stages of infection, during which it primarily functions to inhibit host transcription. The temporal pattern of φCbK *tgrL* transcription is evidence that it functions as an "early" viral gene that supports infection. This study illuminates the functions of a related group of XRE-family TFs that are present in both *Caulobacter* and φCbK, which can influence host adhesin development and promote phage infection.

## Results

### Discovery of putative XRE-family adhesion regulators in *Caulobacter* and its bacteriophage

To assess the conservation and relatedness of known adhesion regulators within the *Caulobacter* genus, we constructed a *Caulobacter* pangenome comprising 19 species isolated from a diverse range of soil and aquatic ecosystems (Fig 2 and S1 Table). Considering the importance of XRE-family transcription factors (TFs) in the regulation of *Caulobacter* holdfast development and adhesion, we primarily focused our analysis on this gene class. We identified a total of 413 XRE domain-containing proteins across the 19 species, which grouped into 57 distinct gene clusters (GCs) (S1 Table). GC_0003, the third largest gene cluster in the pangenome, contained 74 distinct XRE-family TFs, with species encoding up to seven paralogs. Known holdfast regulators RtrA and RtrB [7] and an uncharacterized *C. crescentus* TF encoded by gene locus *CCNA_00049* were contained in GC_0003 (Figs 2 and 3A). Cluster GC_0408 was among the single-copy core genes found in all species and contained StaR, a known regulator of *C. crescentus* stalk development [24] and a repressor of *hfiA* transcription [22] (Figs 2 and 3A). GC_2778 was limited to the *C. vibrioides* and *C. segnis* clades and includes the *C. crescentus* XRE TF gene, *CCNA_02755* (Figs 2 and 3A). Homologs of the unrelated *hfiA* transcriptional repressor, *rtrC* [6], were present in one or two copies in select *Caulobacter* species (GC_2397) (Figs 2 and 3A). Relative protein sequence identity of the nineteen HTH_XRE domain proteins in *C. crescentus* is presented in Fig 3B.

The genomic regions surrounding *rtrA*, *rtrB*, *CCNA_02755*, and *staR* exhibit synteny homology across the *Caulobacter* genus (S1 Fig) while *CCNA_00049* and its adjacent genomic region are conserved across the class *Alphaproteobacteria* (S2 Fig), suggesting that *CCNA_00049* is likely the ancestral XRE-family regulator in *Alphaproteobacteria* [25]. Published RNA-seq and ChIP-seq data on the RtrC adhesion regulator indicate that it directly

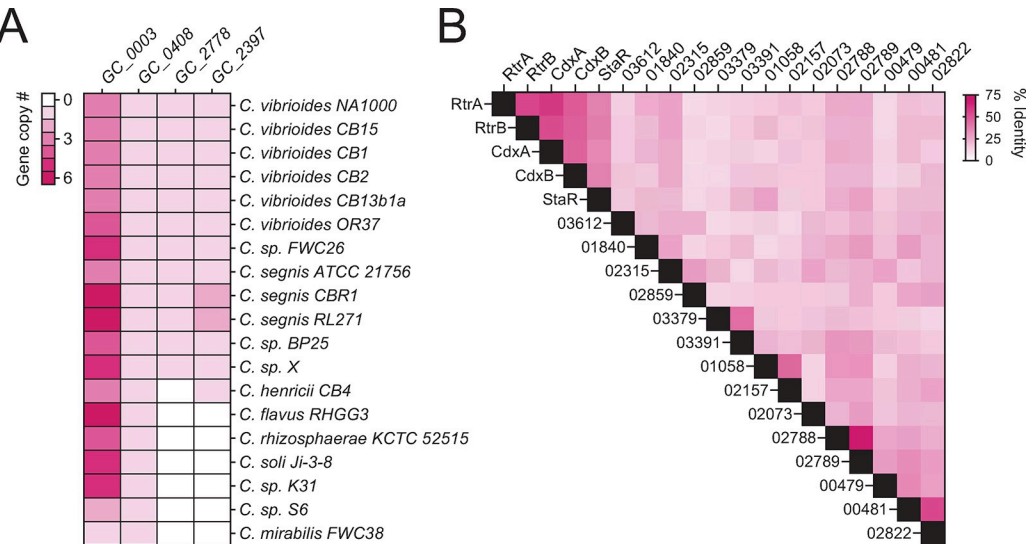

**Fig 3. Pangenome summary of holdfast regulators. A)** Number of paralogs in the gene clusters highlighted in Fig 2 (mcl-inflation = 3). *C. crescentus* members of GC_0003: RtrA, RtrB, CdxA (*CCNA_00049*); GC_0408: StaR; GC_2778: CdxB (CCNA_02755); GC_2397: RtrC. **B)** Percent pairwise amino acid identity between all of the XRE-family transcription factors (HTH_XRE; Conserved Domain Database accession—cd00093) encoded by *C. crescentus* NA1000. Numbers indicate the corresponding locus ID (CCNA_XXXXX).

activates *CCNA_00049* and *CCNA_02755* transcription [6] (S3A–S3D Fig). Consistent with this result, *rtrC* overexpression enhanced *CCNA_00049* and *CCNA_02755* expression by 4.8- and 1.8-fold, respectively in a transcriptional reporter assay (S3B and S3D Fig). We hereafter refer to *CCNA_00049* and *CCNA_02755* as *cdxA* and *cdxB*, for RtrC-dependent XRE family transcriptional regulators, respectively. We conclude that *cdxA* and *cdxB* are XRE-family TFs that function within the recently defined RtrC adhesion regulon [6].

Given the broad conservation *cdxA* in *Alphaproteobacteria*, well-described mechanisms of gene exchange between bacteria and their viruses [26,27], and the established role of XRE-family TFs in bacteriophage biology, we further searched for *cdxA*-related genes in Alphaproteobacterial phage genomes. This effort identified *cdxA*-related genes in φCbK-like Caulophage, *Agrobacterium* phage, *Sinorhizobium* phage, and *Brevundimonas* phage (S4 Fig) suggesting these viral regulatory proteins related to *Caulobacter* adhesion factors impact host processes during phage infection.

## C. crescentus and φCbK XRE proteins repress hfiA expression and promote holdfast production

RtrA, RtrB, and StaR directly repress transcription of the *hfiA* holdfast inhibitor, and thereby promote holdfast development [7,22]. Given the sequence similarity of CdxA and CdxB to these proteins (Figs 2 and 3), we postulated that they would also repress *hfiA* transcription and promote holdfast development. We further predicted that that the CdxA-related regulator from Caulophage φCbK, Gp216, would regulate *Caulobacter* holdfast development. To test these predictions, we overexpressed *C. crescentus* and φCbK XRE TFs, and monitored *hfiA* expression using a fluorescent transcriptional reporter. In M2-xylose defined medium, wild-type cells highly express *hfiA* and, as a consequence, few cells elaborate holdfast in this condition (Fig 4A and 4B). As expected, overexpression of *rtrA* and *rtrB* significantly reduced transcription from the *hfiA* promoter (P$_{hfiA}$) (Fig 4A). Similarly, overexpression of *cdxA*, *cdxB*, or

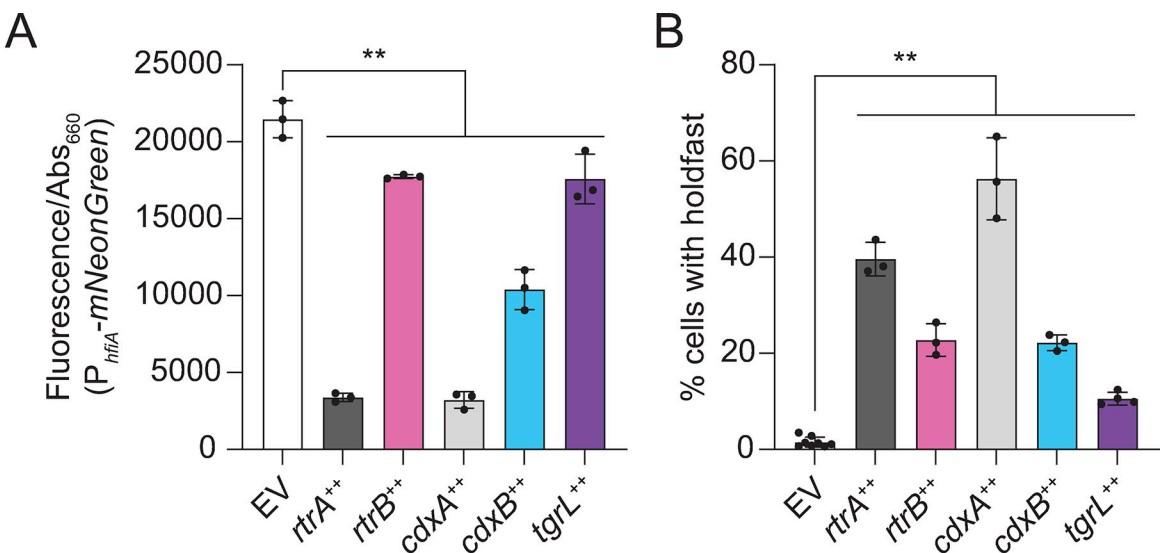

**Fig 4. *C. crescentus* and φCbK XRE regulators repress *hfiA* transcription and promote holdfast development. A)** *hfiA* expression evaluated using a P$_{hfiA}$-*mNeonGreen* transcriptional reporter. Fluorescence was measured in a wild type background containing either the empty vector (EV), or *rtrA*, *rtrB*, *cdxA*, *cdxB*, or φCbK *tgrL* overexpression (++) vectors. Fluorescence was normalized to cell density. **B)** Percentage of cells with stained holdfast. Using the same strains as in (A), stained holdfasts were quantified by fluorescence microscopy. In A and B, cells were grown in M2-xylose defined medium. Bars represent the mean ± standard deviation of at least three biological replicates (dots). Statistical significance was determined by one-way ANOVA followed by Dunnett's multiple comparison (p-value $\leq$ 0.01,**).

φCbK *gp216* reduced P$_{hfiA}$ reporter signal by 85%, 52%, and 18%, respectively (Fig 4A). Consistent with the observed decrease in *hfiA* transcription, inducing expression of *rtrA*, *rtrB*, *cdxA*, *cdxB*, or φCbK *gp216* led to a significant increase in the fraction of cells with holdfasts (Fig 4B). Thus, CdxA and CdxB, like RtrA and RtrB, can promote holdfast development. Additionally, our results indicate that φCbK Gp216, a Caulophage XRE-family regulator, can modulate host holdfast development. We hereafter refer to φCbK *gp216* as φCbK *tgrL* for transcription factor involved in host gene regulation encoded by lytic phage φCbK.

### Functional overlap of C. crescentus and φCbK XRE transcription factors

To define the DNA binding profiles of *C. crescentus* and φCbK XRE TFs, we conducted chromatin immunoprecipitation sequencing (ChIP-seq) with 3x-FLAG-tagged variants of each protein (S2 Table). Consistent with previous work [7], our analysis revealed a significant enrichment of the *hfiA* promoter region upon precipitation of RtrA or RtrB (Fig 5A). ChIP-seq peaks for CdxA, CdxB, and φCbK TgrL exhibited similar enrichment of this promoter (Fig 5A). We comprehensively examined the overlap of ChIP-seq peak summits (+/-50 bp around the peak) across all the datasets for each protein and uncovered considerable binding site overlap: 59% of RtrA (649 out of 1094), 86% of RtrB (468 out of 544), 72% of CdxA (584 out of 809), and 94% of CdxB (191 out of 202) peaks overlapped with at least one other XRE binding site (Fig 5B). Remarkably, 99% of peak summits for the φCbK XRE protein TgrL (135 out of 136) overlapped with the summits from one of the other *Caulobacter* XRE proteins (Fig 5B).

To identify putative binding motifs for these XRE TFs, we analyzed the sequences from the peak summits using the XSTREME tool from the MEME suite [28], and identified similar centralized motifs across all five datasets (Fig 5C). We conclude that that the *C. crescentus* and the φCbK TgrL TFs have largely overlapping binding profiles when overexpressed.

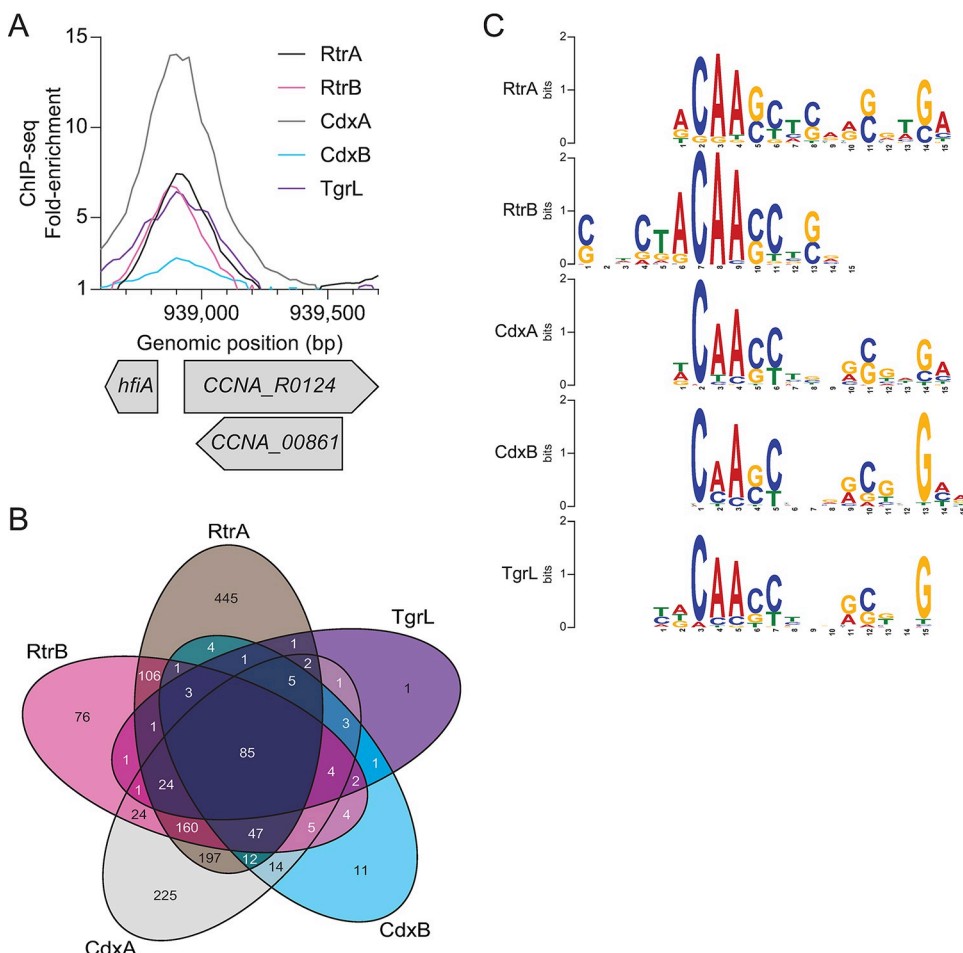

**Fig 5. *C. crescentus* and φCbK XRE regulators have redundant DNA binding repertoires. A)** XRE proteins bind the *hfiA* promoter *in vivo*. ChIP-seq profile from pulldowns of 3xFLAG-tagged XRE proteins are shown. Lines indicate the fold-enrichment from pulldowns compared to an input control. Genomic position and relative position of genes are indicated. Data are in 25 bp bins and are the mean of three biological replicates. **B)** XRE proteins bind overlapping sites on the *C. crescentus* chromosome. Venn diagram showing the number of ChIP-seq peaks (100 bp centered on summit) that occupy overlapping regions as identified by ChIPpeakAnno [83]. **C)** DNA sequence motif enriched in indicated XRE ChIP-seq peaks as identified by XSTREME [82].

## Temporal pattern of φCbK gene expression during infection: tgrL is an early gene

Data presented thus far indicate that φCbK *tgrL* is functionally related to XRE-family adhesion regulators when expression is induced in *C. crescentus* cells, though it was not known if *tgrL* is expressed during φCbK infection. To test this, we conducted time series RNA-sequencing (RNA-seq) measurements on mRNA isolated from infected wild type *C. crescentus* to define the timing and pattern of φCbK gene expression during infection.

By 15 minutes post infection, φCbK mRNA accounted for 14% of total detected transcripts; this fraction rose to 39% by 90 minutes (Fig 6A). Analysis of φCbK gene expression by hierarchical clustering revealed six distinct temporal expression patterns, which we label as early, constitutive, early-middle, middle, late-middle, and late (S5A–S5G Fig). Approximately 41% of φCbK genes reached their peak expression by 15 minutes and then gradually decreased, fitting the early category (S5B Fig). Constitutive genes were maximally expressed by 15 minutes

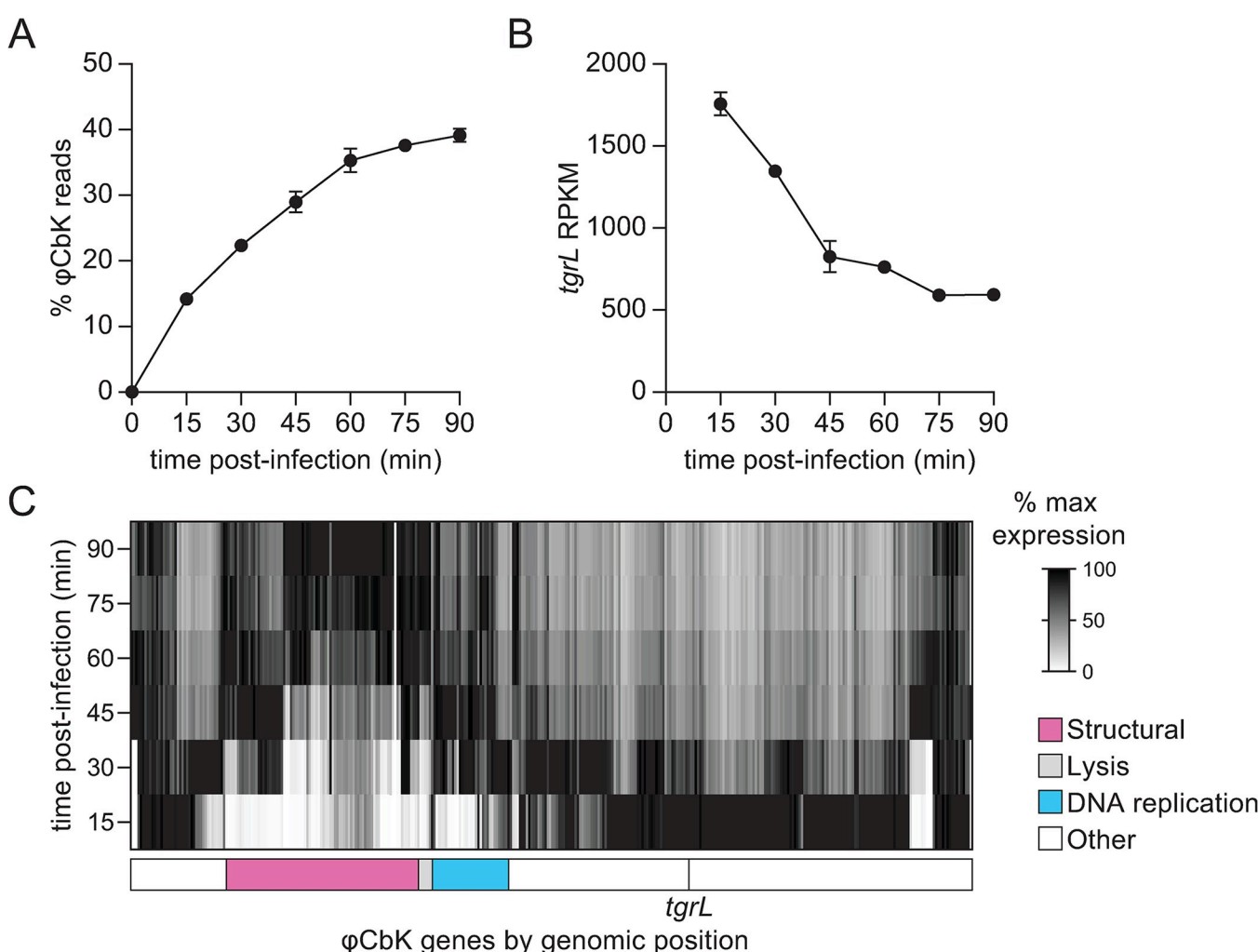

**Fig 6. Temporal expression profile of φCbK transcription during infection of *C. crescentus*. A)** Relative φCbK transcript levels over an infection time course as measured by RNA-seq. Percentage of reads from RNA-seq mapped to the φCbK genome compared to the *C. crescentus* host genome. **B)** φCbK *tgrL* is transcribed at the earliest stages infection. Normalized transcript levels (RPKM) for φCbK *tgrL* from RNA-seq were plotted over the course of an infection. **C)** Relative expression of φCbK genes during an infection. Transcript levels for each φCbK gene were plotted and relative values were calculated by normalizing transcript levels at a time point to the maximum transcript levels for that gene over the infection time course. Columns correspond to φCbK genes and genes are placed in the order that they appear on the phage chromosome. Genomic modules for structural, lysis, and DNA replication genes as indicated by [88] are marked below the heatmap. The position of φCbK *tgrL* is indicated by a vertical line and labeled. **A-C)** Wild type cells were infected during logarithmic growth at 10 multiplicity of infection (MOI). Samples for t = 0 min were harvested prior to phage addition. Data are the mean and error bars represent the standard deviation of three biological replicates.

and maintained this level throughout the infection (S5C Fig). Early-middle and late-middle genes peaked at 30- and 45-minutes post-infection respectively, before diminishing (S5D and S5F Fig), while middle genes achieved their highest expression between 30 and 60 minutes (S5E Fig). Late genes steadily rose to maximum levels between 60- and 90-minutes post-infection (S5G Fig). The complete φCbK temporal expression profile is presented in S3 Table.

φCbK *tgrL* was highly expressed and clearly clustered into the early group, peaking at 15 minutes post-infection and declining thereafter (Fig 6B and 6C). Steady-state transcript levels of φCbK *tgrL* were 6 to 28 times higher than its host homologs, *rtrA*, *rtrB*, *cdxA*, and *cdxB* by 15 minutes. Overall, temporal patterns of bacteriophage gene expression were as expected. For example, DNA replication genes within the φCbK genome were expressed early in infection,

while structural and lysis genes were expressed in the middle and late stages of infection (Fig 6C and S3 Table).

## Regulation of transcription by *Caulobacter* and φCbK XRE proteins

To determine how the *C. crescentus* XRE TFs impacted gene expression, we measured transcript levels in a wild type and Δ*rtrA* Δ*rtrB* Δ*cdxA* Δ*cdxB* (i.e. quadruple XRE deletion) backgrounds by RNA-seq. Transcripts of 112 genes differed significantly between the two backgrounds (S4 Table). Direct targets of the *C. crescentus* XRE TFs were defined as genes that *a)* showed significant differential regulation in the quadruple deletion background (|fold-change| $\geq$ 1.5 and FDR p-value $\leq$ 0.0001), and *b)* had a ChIP-seq peak for at least one of the *C. crescentus* XRE TFs present within an associated promoter (S4 Table). In total, 79 genes were defined as direct targets and 94% (74 out of 79) were upregulated in the Δ*rtrA* Δ*rtrB* Δ*cdxA* Δ*cdxB* strain (S4 Table). These results indicate that this group of *C. crescentus* XRE TFs primarily function as transcriptional repressors in their host system. Among the functional classes of genes significantly upregulated in the quadruple XRE deletion background were toxin-antitoxin (TA) and nuclease genes, which have been implicated in host-phage interactions in some systems [29]. For example, *CCNA_03255* (PemK-like toxin) and *CCNA_03983* (HicA-like toxin) increased 2- and 4-fold respectively upon XRE deletion (S4 Table), while levels of the GIY-YIG nucleases *CCNA_00744* and *CCNA_01405* increased 11- and 2-fold, respectively (S4 Table).

To define the impact of φCbK *tgrL* on transcription of *C. crescentus* genes, we used multiple approaches. Overexpressing φCbK *tgrL* in the quadruple deletion background (Δ*rtrA* Δ*rtrB* Δ*cdxA* Δ*cdxB*) allowed us to determine a TgrL regulon that was not confounded by expression of closely related host XRE TFs. Transcript levels of 440 genes changed significantly when φCbK *tgrL* was overexpressed (Fig 7). Direct targets of this early φCbK TF were defined as genes that *a)* showed significant differential regulation upon φCbK *tgrL* overexpression (|fold-change| $\geq$ 1.5 and FDR p-value $\leq$ 0.0001), and *b)* had a TgrL ChIP-seq peak present within an associated promoter (S4 Table). In total, 99 regulated genes were defined as direct TgrL targets. Among these, 93% (92 out of 99) were repressed (Fig 7) including the holdfast inhibitor, *hfiA*, which decreased by a factor of approximately six. These results provide evidence that φCbK *tgrL* can act as a direct regulator of *C. crescentus* gene expression, including many genes that have direct binding sites for homologous chromosomally-encoded XRE adhesion regulators (S4 Table).

We further analyzed our RNA-seq infection time course data to define changes in host transcription, focusing on genes that were classified as direct targets of φCbK TgrL. Transcription of 337 *C. crescentus* genes changed significantly relative to pre-infection at one or more time points, and differentially expressed genes clustered into distinct temporal groups. The largest cluster (Group A) contained 120 genes that had maximum transcript levels prior to infection, which decreased following infection. Multiple group A genes contained φCbK TgrL ChIP-seq peaks within their promoter regions. This cohort includes the GIY-YIG nuclease genes presented above (*CCNA_00744* and *CCNA_1405*); the *cdzAB* transporter genes implicated in contact-dependent toxin transport; an operon encompassing the *lodAB*-like L-amino acid oxidase genes (*CCNA_00592–00589*); a pair of hypothetical protein genes (*CCNA_00593–00594*); and the operon *CCNA_02815–02816*, encoding an ice nucleation protein and a hypothetical protein. Though not likely regulated directly by φCbK TgrL, the transcription of a subset of oxidative stress defense genes including the catalase-peroxidase *katG* (*CCNA_03138*), alkyl hydroperoxide reductase subunits C and F (*CCNA_03012* and *CCNA_03013*, respectively), and an AhpD-family hydroperoxidase (*CCNA_03812*), were markedly repressed post-

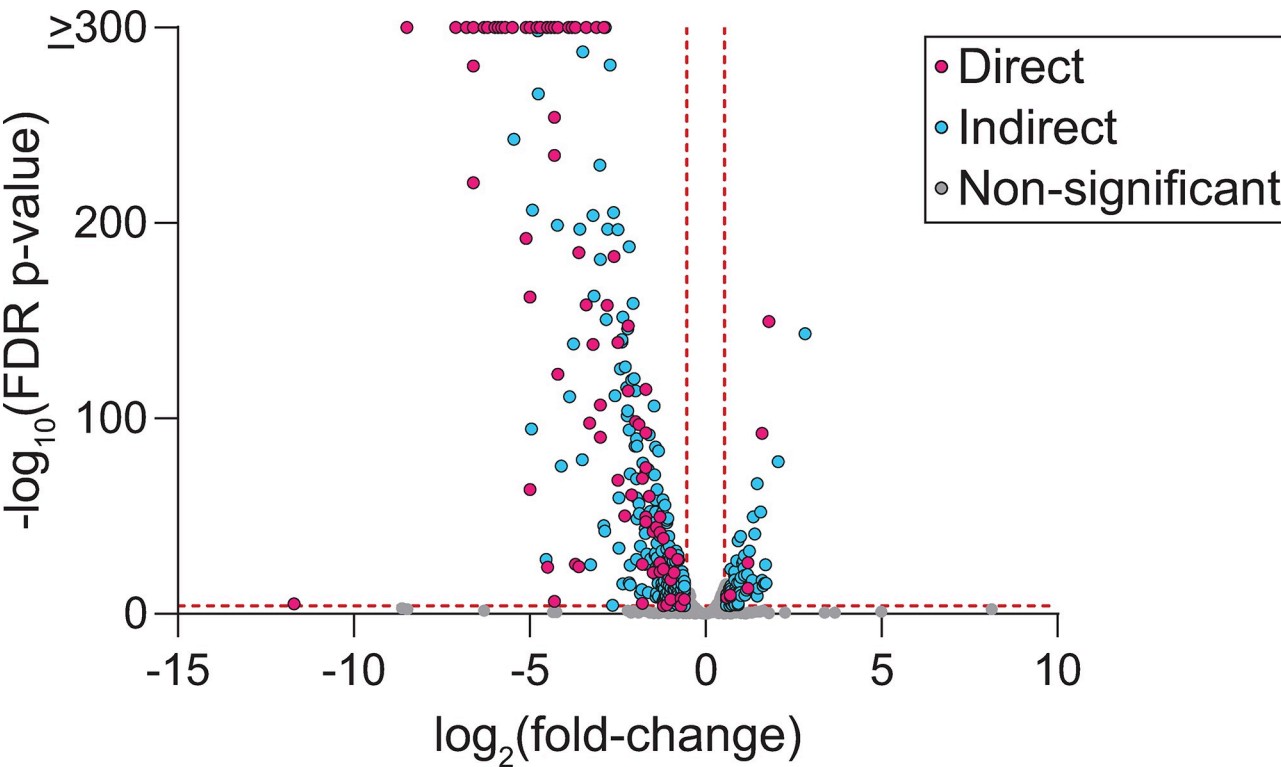

**Fig 7. TgrL regulates *C. crescentus* gene expression.** RNA-seq analysis of genes significantly regulated upon induction of φCbK *tgrL* expression. Volcano plot displays log$_2$(fold-change) and -log$_{10}$(FDR p-value) for all *C. crescentus* genes, comparing φCbK *tgrL* expression from a plasmid to that from an empty vector (EV). Experiment was conducted in a genetic background lacking the host XRE genes: Δ*rtrA* Δ*rtrB* Δ*cdxA* Δ*cdxB* background. Gray dots indicate genes for which expression does not change significantly, blue dots indicate genes without a TgrL ChIP-seq peak in an associated promoter, and pink dots indicate genes with a φCbK TgrL ChIP-seq in an associated promoter. Vertical red lines mark boundaries for ±1.5-fold-change; horizontal red line marks 0.0001 FDR p-value. Points represent the mean of three biological replicates.

infection (S3 Table) suggesting that φCbK infection represses oxidative stress response. The impact of this response on host and/or phage fitness is uncertain.

### *Caulobacter* and φCbK XRE proteins repress cell lysis activators

To better understand the functions of both host and phage XRE TFs, we looked for genes that had common XRE TF ChIP-seq peaks in their promoter regions. Notably, we identified ChIP-seq peaks for RtrA, RtrB, CdxA, CdxB, and φCbK TgrL directly upstream of the *gafYZ* operon (Fig 8A). Expression of *gafYZ* induces expression of prophage-like gene transfer agents (GTA) that package genomic DNA fragments from *C. crescentus* and trigger cell lysis [23]. Induction of host cell lysis prior to the completion of a phage infection cycle would be deleterious for phage fitness, and we hypothesized that φCbK TgrL represses GTA expression. To test this hypothesis, we measured *gafYZ* transcriptional reporter activity in a wild type background and in a quadruple deletion background (Δ*rtrA* Δ*rtrB* Δ*cdxA* Δ*cdxB*), which lacks the related host XRE TFs. Deletion of the four host XRE TFs resulted in a 2.3-fold increase in *gafYZ* transcription (Fig 8B). Expressing *rtrA*, *rtrB*, *cdxA*, *cdxB*, or φCbK *tgrL* in the quadruple deletion background reduced *gafYZ* transcription by 34–91% (Fig 8B).

Prior research [23] has demonstrated that deleting *rogA*, a strong repressor of *gafYZ*, stimulates the production of GTAs during stationary phase and results in packaging of chromosomal DNA into 8 kb fragments. To assess production of GTA-associated DNA fragments in

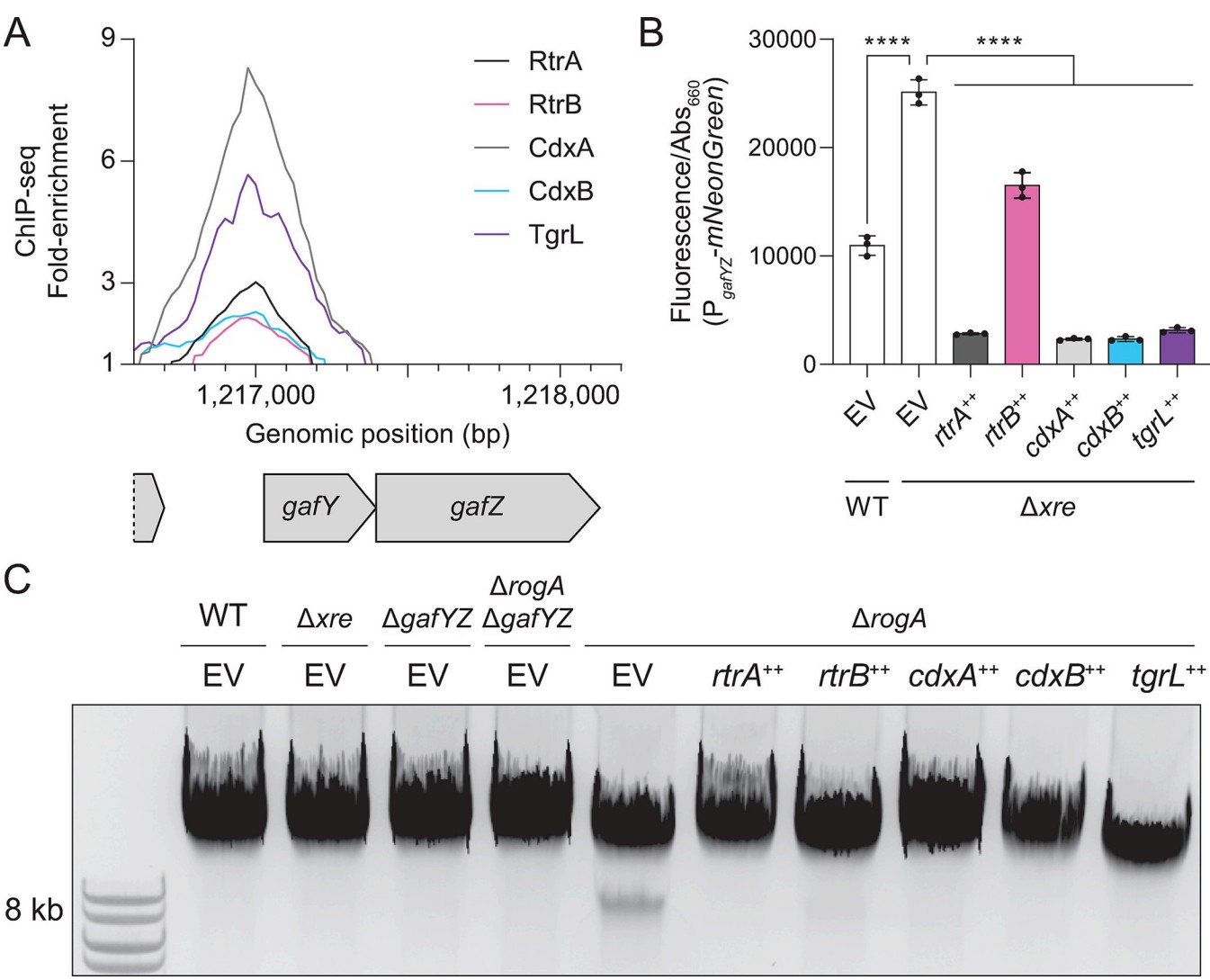

**Fig 8. *C. crescentus* and φCbK XRE regulators repress *gafYZ* and gene transfer agent (GTA) production. A)** XRE proteins bind the *gafYZ* promoter *in vivo*. ChIP-seq profile from pulldowns of 3xFLAG-tagged protein are shown. Lines show fold-enrichment from pulldowns compared to an input control. Genomic position and relative position of genes are indicated. Data are in 25 bp bins and are the mean of three biological replicates. **B)** *gafYZ* expression using a $P_{gafYZ}$-*mNeonGreen* reporter. Fluorescence was measured and normalized to cell density. Data are the mean and error bars are the standard deviation of three biological replicates (black dots). Statistical significance was determined by one-way ANOVA followed by Šídák's multiple comparisons test (p-value $\leq$ 0.0001, ****). **C)** XRE proteins repress GTA production. Total DNA was purified, separated by gel electrophoresis, and imaged. GTA-associated DNA resolved at ~8 kb. Image is a representative gel from at least 3 biological replicates. **B-C)** Experiments were performed with wild type (WT) or strains harboring in-frame deletions (Δ) in *rtrA*, *rtrB*, *cdxA*, *cdxB*, *gafYZ*, and/or *rogA*. Δ*xre* indicates the Δ*rtrA* Δ*rtrB* Δ*cdxA* Δ*cdxB* background. Strains contained either an empty vector (EV), *rtrA*, *rtrB*, *cdxA*, *cdxB*, or φCbK *tgrL* overexpression (++) vectors. Strains were grown to stationary phase in complex medium (PYE).

the quadruple deletion, we extracted DNA from stationary phase cultures and performed gel electrophoresis. As expected, we observed an 8 kb GTA-associated DNA band in Δ*rogA*, but not in a Δ*rogA* Δ*gafYZ* mutant (Fig 8C). However, increased transcription of *gafYZ* in the quadruple deletion was not sufficient to trigger the production of GTA fragments. Although we failed to detect a GTA-associated DNA band in the quadruple deletion, expression of either *rtrA*, *rtrB*, *cdxA*, *cdxB*, or φCbK *tgrL* was sufficient to ablate production of the 8 kb GTA band in a Δ*rogA* strain (Fig 8C). These observations support a model in which related XRE TFs from both *C. crescentus* and φCbK can repress GTA production.

### *Caulobacter* and φCbK XRE transcription factors promote phage infection

A transposon-based genetic screen previously revealed that resistance to φCbK phage infection was increased in mutants harboring insertions in *cdxB*, which indicated a role for *cdxB* in supporting phage infection [30]. Consistent with this published Tn-seq result, we observed that phage burst size was significantly reduced (by 41%) in a Δ*cdxB* strain (Fig 9A). Given the functional overlap of *cdxB* with other XRE proteins described in this study, we postulated that the other host (*rtrA*, *rtrB*, *cdxA*) and φCbK (φCbK *tgrL*) XRE TFs are genetic factors that support φCbK infection. We assessed viral burst size in the *C. crescentus* quadruple deletion strain (Δ*rtrA* Δ*rtrB* Δ*cdxA* Δ*cdxB*) and in strains where each XRE TF was individually expressed. Deletion of all four paralogous host XRE TFs reduced burst size to a greater extent than *cdxB* deletion alone, though the difference between Δ*cdxB* and the quadruple XRE deletion strain was not statistically significant suggesting that *cdxB* is the major contributor to this burst size phenotype (Fig 9A). Nonetheless, expression of either *rtrB*, *cdxA*, *cdxB*, or φCbK *tgrL* was sufficient to rescue the burst size defect of the quadruple XRE deletion strain (Fig 9B) providing evidence that these related XRE TFs from both the host and virus can support φCbK phage infection.

### *Caulobacter* and φCbK XRE transcription factors can form heteromers

Transcription factors within the XRE family are known to form homomeric structures, such as homodimers or homotetramers [31–34]. Given the structural similarity among RtrA, RtrB, CdxA, CdxB, StaR, and φCbK TgrL XREs, we postulated that these proteins could form heteromeric interactions, in addition to homomeric associations. To test this hypothesis, we employed a bacterial two-hybrid (BTH) assay, wherein each XRE was fused to the C-terminus of either the T18 or T25 fragments of adenylate cyclase. As expected, co-expression of any

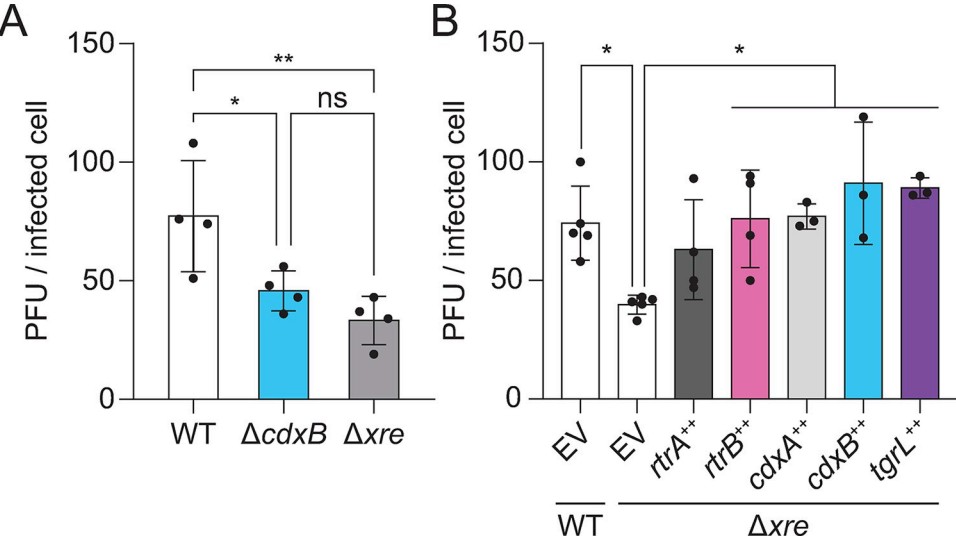

**Fig 9. *C. crescentus* and φCbK XRE regulators support phage infection.** φCbK burst size in mutant backgrounds. **A-B)** Plaque forming units (PFU) per infected cell (i.e. burst size) was plotted for either wild type or mutants with in-frame deletions (Δ) of *rtrA*, *rtrB*, *cdxA*, and/or *cdxB*. Δ*xre* indicates the Δ*rtrA* Δ*rtrB* Δ*cdxA* Δ*cdxB* quadruple deletion background. **B)** Strains contained either an empty vector (EV), or *rtrA*, *rtrB*, *cdxA*, *cdxB*, or φCbK *tgrL* overexpression (++) vectors. **A-B)** Strains were infected with φCbK at 0.01 multiplicity of infection (MOI) in logarithmic growth phase in complex medium (PYE). Data bars represent the mean and error bars are the standard deviation of at least three biological replicates (black dots). Statistical significance was determined by one-way ANOVA followed by Dunnett's multiple comparison (p-value ≤ 0.05,*; ≤ 0.01,**).

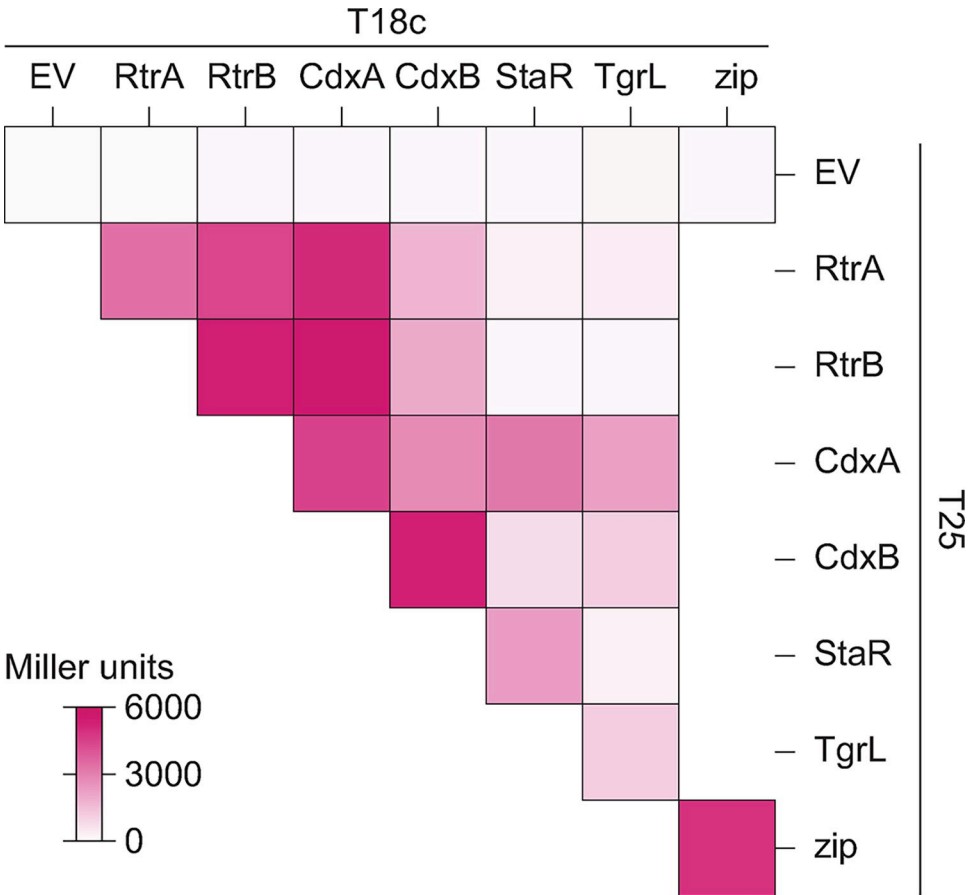

**Fig 10.** *C. crescentus* **and φCbK XRE regulator interactions in a two-hybrid assay.** Heatmap summarizing interactions between RtrA, RtrB, CdxA, CdxB, StaR, and φCbK TgrL based on bacterial two-hybrid (BTH) assays. Proteins were fused to split adenylate cyclase fragments (T18c and T25) and co-expressed in *E. coli*. Interactions between the fused proteins reconstitutes adenylate cyclase, promoting expression of a *lacZ* reporter. Empty vector (EV) are the negative control and Zip is the positive control. β-galactosidase activity was measured for each pairing, and Miller units were calculated. Data are the mean of at least three biological replicates. See S6 Fig for statistical analysis.

XRE gene with itself resulted in increased β-galactosidase activity, demonstrating that the five *C. crescentus* XRE proteins and the phage XRE protein form homomeric structures (Fig 10). We then co-expressed different XRE proteins and discovered that RtrA, RtrB, CdxA, and CdxB interacted in a two-hybrid assay (Fig 10). StaR and φCbK TgrL interacted with CdxA and CdxB, but not with RtrA or RtrB (Figs 10 and S6). In fact, interaction of φCbK TgrL with itself was as strong as its interactions with CdxA and CdxB in this assay. The ability of φCbK TgrL to interact with both CdxA and CdxB points toward a possible mechanism in which this phage protein can impact host gene expression directly or indirectly by interacting with host transcription factors.

## Discussion

A pangenome analysis of the *Caulobacter* genus illuminated a probable evolutionary radiation of a group of XRE-family transcription factors (XRE TFs) that includes several genes previously implicated in the regulation of cell adhesion. Study of this group of XRE TFs in *C. crescentus* and of a closely related gene encoded by Caulophage φCbK (*tgrL*) has illuminated new functional roles for this gene class in *Caulobacter*-virus interactions. Expression of φCbK *tgrL*

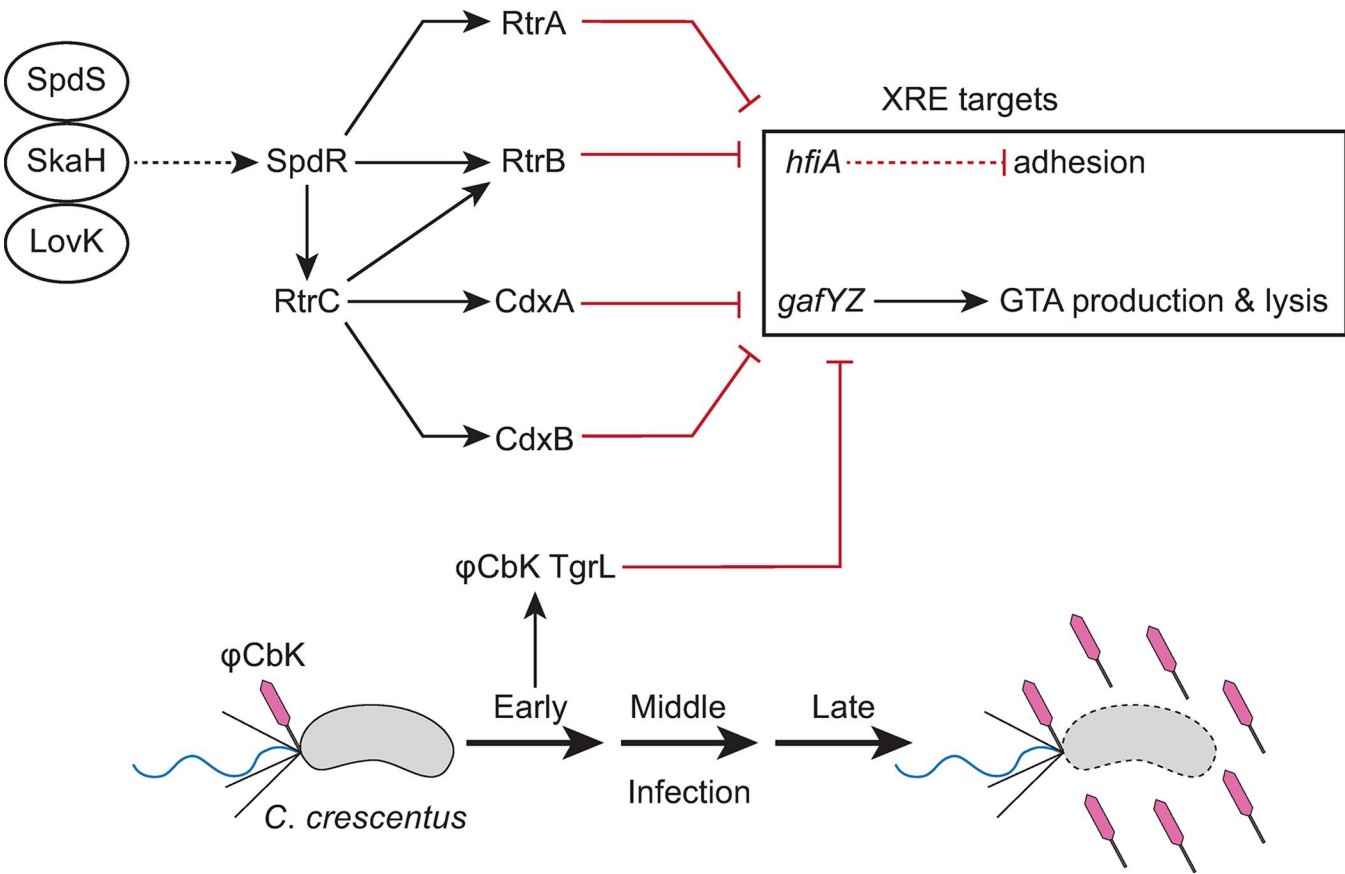

**Fig 11. XRE Transcription Factor Network in regulation of *Caulobacter* and φCbK gene expression.** Schematic of XRE transcription factor (TF) network (top) and φCbK infection schematic (bottom). The sensor histidine kinases LovK, SkaH, and SpdS physically interact and promotes SpdR activity [7]. SpdR activates *rtrC* expression [6], and RtrC and SpdR activate expression of XRE TF paralogs (*rtrA*, *rtrB*, *cdxA*, and *cdxB*) [6, 7]. XRE TFs repress holdfast and gene transfer agent (GTA) regulators, *hfiA* and *gafYZ*, respectively. φCbK *tgrL* is highly expressed during early infection of *C. crescentus* and regulates expression of *C. crescentus* genes. Dashed lines indicate post-transcriptional regulation and solid lines indicate transcriptional regulation. Black arrows indicate activation and red bar-ended lines indicate repression. Pink phage indicates φCbK phage, grey cells indicate *C. crescentus*, and cells outlined in dashed lines indicates lysed cells.

is highly induced in the early stages of infection of host cells, i.e. *tgrL* is an "early" gene. The transcriptional regulon and functional properties of this viral transcription factor overlap its host-encoded homologs. These TFs can not only govern cell adhesion processes, but also repress transcription of a regulator associated with the gene transfer agent (GTA) locus, which produces bacteriophage-like particles that encapsulate host DNA (Fig 11). Our data further provide evidence that both host and φCbK XRE TFs play an important role in promoting φCbK infection. Specifically, deletion of related host XRE TFs led to a reduction in viral burst size, while expression of these XRE TFs was sufficient to rescue this burst size defect.

## An ancestral XRE TF and its function across genera

The gene neighborhoods of the XRE TFs studied here are highly conserved in *Caulobacter*. Synteny homology of *cdxA* (*C. crescentus* gene locus: *CCNA_00049*) is particularly notable as its neighborhood is conserved across the class Alphaproteobacteria (S2 Fig) [25], suggesting it is the ancestral XRE TF in *C. crescentus*. Orthologs of this gene have been studied in several genera and demonstrated to regulate diverse processes. For example, mutation of the *Azorhizobium caulinodans cdxA* homolog, *praR*, resulted in abnormal root nodule development in its

plant host and reduced nitrogen fixation during symbiosis [25,35]. Anomalous nodulation of *praR* mutants was attributed to increased expression of the *reb* locus, known to control the formation of R-bodies [25], which can modulate interactions between bacteria and eukaryotic cells [36]. The disruption of *praR* in *Rhizobium leguminosarum bv. viciae* 3841 resulted in enhanced adhesin production, which is opposite of the adhesion phenotypes that we have reported in *C. crescentus*. Increased adhesin production in the *Rhizobium* system is a consequence of the interaction of the PraR repressor with CinS, an anti-repressor [37] that facilitates synchronization of biofilm regulation with population density [37,38] and improves fitness in the root nodule [38]. Notably, *R. leguminosarum* encodes a *praR* paralog, *RL0149*, that is repressed by PraR but does not impact biofilm formation [38]. However, *RL0149* mutants face a competitive disadvantage in the root nodule, providing evidence for functional diversification of these XRE TF paralogs in this species [39].

Homologs of XRE TFs studied here were also identified in several Alphaproteobacterial phages, including *Caulobacter* phage φCbK, *Brevundimonas* phage vB_BgoS-Bajun, *Sinorhizobium* phage PBC5, and *Agrobacterium* phage Atu_ph08 (S4 Fig). We provide evidence that the XRE homologs from *C. crescentus* and its phage, φCbK, support phage infection. It is unclear whether this role in phage infection is conserved in other *Alphaproteobacteria* and their respective phages. Considering that *cdxA* and its orthologs can regulate diverse processes in *Alphaproteobacteria*, it is possible that the regulatory functions of phage XRE homologs vary from host to host.

## On the regulation of XRE-family transcription factors

While we have characterized the targets and transcriptional outputs of a related set of XRE-family TFs in *C. crescentus* and φCbK, there is still much to learn regarding mechanisms that regulate their activities. In many systems, XRE-family TFs are controlled through post-transcriptional mechanisms. Among the most studied XRE TFs are the Cro and CI proteins from bacteriophage 434 and λ. In phage λ, the CI protein contains a DNA-binding N-terminal HTH_XRE domain and a C-terminal LexA-like S24 peptidase domain promoting oligomerization [40]. Activation of the host SOS response induces CI self-cleavage between these domains, leading to CI inactivation. This inactivation suppresses lytic-pathway genes, promoting phage induction [41]. However, the XRE-family TFs in *C. crescentus* and φCbK lack a recognizable C-terminal domain, indicating they likely aren't regulated by a related mechanism.

Other XRE-family TFs, such as *Bacillus subtilis* SinR, are regulated by protein-protein interactions. SinR activity is modulated post-transcriptionally by its interaction with the SinI anti-repressor. This interaction disrupts SinR DNA binding and multimerization, depressing SinR targets [34,42,43]. Similar regulatory interactions are seen with the CdxA ortholog from *R. leguminosarum bv. viciae* 3841, known as PraR [37]. Here, the anti-repressor CinS interferes with PraR DNA binding, relieving transcriptional repression. [37]. A BLAST search failed to reveal a CinS-like protein in *C. crescentus*, suggesting there may be undiscovered, distinct anti-repressors. Indeed, observed protein-protein interactions between these XRE paralogs (Fig 10) may have anti-repressive functions.

Another possibility is that the XRE TFs from *C. crescentus* and φCbK are constitutively active and are primarily regulated through transcription. Expression of the *cdxA* ortholog from *Sinorhizobium medicae*, *phrR*, is upregulated in a variety of stress conditions, including low pH, high concentrations of $Cu^{2+}$ or $Zn^{2+}$, ethanol stress, and hydrogen peroxide stress [44]. In contrast, expression of *praR*, the *cdxA* ortholog from *A. caulinodans*, is not induced under low pH conditions, suggesting that the regulatory inputs of these conserved TFs vary across Alphaproteobacteria. In *C. crescentus*, it is established that expression of these XRE TFs

is modulated by SpdR-RtrC pathway, which responds to changes in media conditions and growth phase [45].

## *Caulobacter* and phage XRE proteins add to an expanding group of holdfast regulators

The process of surface attachment in *C. crescentus*, which is mediated by the adhesin known as the holdfast, is a permanent event and is therefore subject to intensive transcriptional and post-transcriptional regulation. The small protein HfiA plays a crucial role in regulating holdfast synthesis by binding and inhibiting HfsJ, a membrane-associated glycosyltransferase that is essential for holdfast production [22]. A complex array of regulatory factors coordinately control *hfiA* transcription [6,7,22,46,47] including an assortment of transcription factors that are activated by the stationary phase response regulator, SpdR [7]

We have discovered two new XRE family transcription factors, CdxA and CdxB. These proteins function within the *C. crescentus* adhesion control system to repress *hfiA* and enhance holdfast synthesis (Figs S3A–S3D, 4A, and 4B), raising the total number of direct *hfiA* regulators to eight. The advantage of the highly redundant, and seemingly baroque, regulation of *hfiA* certainly piques curiosity. A possible explanation could be that regulatory redundancy buffers the cell against transient environmental fluctuations that impact holdfast development. Alternatively, this array of transcription factors could allow for a multi-dimensional response to varying environmental cues that impact *hfiA* transcription. Although the SpdR-RtrC signaling axis is known to activate the quartet of XRE genes studied here (*rtrA*, *rtrB*, *cdxA*, and *cdxB*), it remains uncertain whether other regulatory systems or signals influence their expression or activity (as discussed in the section above). If secondary mechanisms—such as protein-protein interactions, chemical modifications, or metabolite binding—further regulate the XRE proteins, they could introduce an added layer of complexity to the adhesion decision circuit.

Intriguingly, we discovered that a related Caulophage XRE gene, φCbK *tgrL*, can directly repress *hfiA* transcription and enhance holdfast formation (Figs 4 and 11). This raises questions about potential advantages of surface adherence regulation by Caulophage. For example, might phage promote holdfast formation under select conditions to facilitate spread of phage progeny in surface-associated biofilms? Given that the infection strategy of φCbK requires the presence of flagella and pili on the cell surface, which only exist in swarmer and pre-divisional cells [48–50], such a tactic could offer competitive benefits. Alternatively, possible phage-induced modulation of *hfiA* during infection might not directly boost phage fitness and could merely be an incidental consequence of the phage harnessing host XRE TFs to regulate other processes. Indeed, we did not observe significant regulation of *hfiA* in *C. crescentus* cells infected with φCbK under our experimental conditions (S3 Table), though *hfiA* is highly sensitive to nutritional changes and infections were performed under high peptone conditions where *hfiA* expression is very low [22]. Examining the impact of phage infection on holdfast production under different nutritional conditions will better define this regulatory connection.

### Functional redundancy and heteromeric interactions

Transcription factors with shared ancestry often bind similar DNA motifs due to highly conserved DNA-binding domains. In line with this, we found that 59–99% of binding sites identified in the RtrA, RtrB, CdxA, CdxB, and φCbK TgrL ChIP-seq datasets exhibit overlap with each other (Fig 5B). Functional redundancy can allow for the evolution of ecoparalogs, in which paralogous proteins perform the same function but are expressed under different environmental conditions [51]. For the *C. crescentus* XRE TFs studied here, all four are controlled

by the same transcriptional regulatory pathway. From this we infer that these genes are typically expressed under similar conditions (S3A–S3D Fig) [6, 7, 52] though we cannot exclude the possibility that additional transcriptional or post-transcriptional mechanisms modulate their expression or activities.

Paralogs can also undergo subfunctionalization, where the function of the original gene is partitioned, or neofunctionalization, in which a completely new function evolves. [53]. The ChIP-seq data revealed that a portion of peaks (41% for *rtrA*, 14% for *rtrB*, 28% for *cdxA*, and 6% for *cdxB*) are unique, suggesting a degree of functional specialization among the *C. crescentus* XRE paralogs (Fig 5B). RtrA, RtrB, CdxA, CdxB, and φCbK TgrL can form both homomers and heteromers in a two-hybrid assay (Fig 10), and it is possible that these proteins bind to different sites depending on the levels of their interaction partners. Gene expression regulation often leverages heteromerization. For instance, the *Escherichia coli* response regulator RcsB is known to form direct interactions with a variety of TFs to control disparate gene expression processes [54–56]. In *C. crescentus* the paralogous zinc-finger transcription factors, MucR1 and MucR2, known for their roles in cell cycle control, can form heterodimers [57]. Finally, in *Streptomyces venezuelae*, BldM homodimers control the activation of early developmental genes, whereas heterodimers of BldM and WhiI regulate a group of late-stage developmental genes [58]. These dimers display distinct DNA binding patterns: BldM homodimers bind palindromic sequences with two BldM half-sites, whereas BldM-WhiI heterodimers bind non-palindromic sequences featuring one half-site each for BldM and WhiI [58].

MEME analysis of each XRE TF from *C. crescentus* in our study revealed a prevalent motif with one highly conserved half-site and a second, weakly conserved half-site (Fig 5C). Considering the sequence identity between the DNA binding domains of the XRE proteins (64–80% identity), it is plausible that each exhibit subtle differences in DNA binding preference. The weakly conserved second half-site could stem from mixed signals of homomers and heteromers in the ChIP-seq data. The formation of heteromers might not only alter binding site specificity but could also influence the regulatory functions of these XRE TFs.

### *Caulobacter* and phage XREs repress GTA production

Expression of a prophage-like gene transfer agent (GTA) in *C. crescentus* results in packaging of genomic DNA into phage-like particles that ultimately lead to cellular lysis [23]. To prevent genome packaging and untimely lysis, the expression of the GTA gene cluster is tightly regulated: transcription of GTA genes is activated by GafYZ while RogA represses *gafYZ* expression [23]. Our study provides evidence that RtrA, RtrB, CdxA, and CdxB directly repress the transcription of *gafYZ* (Fig 8A and 8B), though deletion of these four XRE genes did not result in increased GTA-related DNA production (Fig 8C). Therefore, RogA is likely the dominant inhibitor of *gafYZ*, while the Rtr and Cdx proteins may function to modulate *gafYZ* expression under particular environmental conditions. *C. crescentus* GTA synthesis has only been reported in stationary phase cultures, which indicates that there are intra- or extracellular cues present in stationary phase that enhance the expression of *gafYZ* through an unknown mechanism [23]. *rtrA* and *rtrB* may contribute to stationary phase control of *gafYZ* as expression of these genes increases during stationary phase as a consequence of increased activity of the stationary phase response regulator, SpdR [6, 52].

We have further discovered that φCbK TgrL can directly repress *gafYZ* expression and inhibit GTA production (Figs 8B–8C and 11). Many bacteria are equipped with abortive infection (Abi) modules that detect phage infection and prompt cell death before replication is complete. This mechanism effectively prevents viral spread within the population [59]. While a phage-sensing module hasn't been identified for the *gafYZ* module, it is possible that the

production of GTA and subsequent cell lysis during phage replication could stem the spread of infection, much like Abi. We propose that φCbK synthesizes φCbK TgrL in part to ensure strong repression of *gafYZ* transcription, thereby blocking GTA production. Our infection time course data align with this hypothesis, as the levels of φCbK *tgrL* transcripts peak at the earliest stages of infection. Despite an observed increase in φCbK *tgrL* expression, we did not observe changes in the levels of *gafYZ* transcripts in our experiment. This could be due to several factors. First, under the conditions tested, the host repressor proteins (RogA, RtrA, RtrB, CdxA, and CdxB) may already maximally inhibit *gafYZ* expression. Alternatively, φCbK TgrL could repress *gafYZ* expression at time points earlier than we have measured. While a phage sensing module tied to GTA activation has not been identified, *gafYZ* expression might be triggered during infection when φCbK *tgrL* is absent. Phage strains lacking φCbK *tgrL* would enable tests of this model.

## φCbK infection and XRE transcription factors

Transposon insertions in *cdxB* increased host resistance to the φCbK phage [30]. This is consistent with our results showing that deleting four XRE family TFs or *cdxB* alone reduced the phage burst size and supports our conclusion that these TFs support phage replication (Fig 9A and 9B). Ectopic expression of φCbK *tgrL*, *rtrB*, *cdxA*, or *cdxB* was sufficient to restore burst size to wild type levels in a quadruple deletion background (Δ*rtrA* Δ*rtrB* Δ*cdxA* Δ*cdxB*). Phage infections conducted in this study used a wild type φCbK phage, which expresses φCbK TgrL at high levels in the early stages of infection. Accordingly, the phage resistance phenotype of the quadruple deletion strain may be more pronounced if one were to infect *C. crescentus* with a φCbK strain lacking *tgrL*.

The mechanisms by which XRE TFs from *C. crescentus* enhance φCbK infection remain undefined. It is possible that the XRE deletion strains studied here are 'primed' against phage infection due to upregulation of protective gene(s) within the XRE TF regulon. For example, numerous toxin-antitoxin (TA) systems are known to function as phage defense genes [29] and we observed upregulation of TA genes in the Δ*rtrA* Δ*rtrB* Δ*cdxA* Δ*cdxB* background, including *CCNA_03255* (PemK-like toxin) and *CCNA_03983* (HicA-like toxin) (S4 Table). Additionally, GIY-YIG nuclease proteins (*CCNA_00744* and *CCNA_01405*), which are typically found in restriction modification systems and homing endonucleases [60,61], were elevated 11- and 2-fold respectively in the quadruple XRE deletion strain (S4 Table). Recent studies have shown that a GIY-YIG-based system can protect *E. coli* from T4 phage infection [62] while a chimeric GIY-YIG protein provides the ICP1 phage with immunity against parasitic mobile genetic elements during infection of *Vibrio cholerae* [63]. Elevated expression of these GIY-YIG genes in *C. crescentus* may provide an advantage for the host during the infection process.

We hypothesize that early expression of φCbK *tgrL* during infection allows the phage to hijack the host XRE transcriptional program, thereby boosting phage production. Supporting this model, overexpressing φCbK *tgrL* in the Δ*rtrA* Δ*rtrB* Δ*cdxA* Δ*cdxB* (i.e. quadruple deletion) background significantly altered the expression of 92% of genes (103 out of 112) that were differentially expressed in the quadruple deletion relative to wild type. Among the host genes that were repressed by inducing φCbK *tgrL* were the TA genes, *CCNA_03255* and *CCNA_03983* (by approximately a factor of 10) and the GIY-YIG genes *CCNA_00744* and *CCNA_01405* (by 50- and 3-fold, respectively) (S4 Table). *Bacillus* phages have also been reported to co-opt host gene regulatory networks by expressing host-related sigma factors that modify host transcription processes, impacting phage fitness and sporulation [64]. Leveraging host-derived regulatory proteins to influence phage infection offers intriguing prospects for the development of engineered recombinant phages.

Another mechanism by which this group of *C. crescentus* XRE TFs may influence production of bacteriophage is by directly controlling the transcription of φCbK genes, as observed in other host-phage systems. For example, the expression of both early and late φ29 genes is downregulated by *spo0A*, a master regulator of *B. subtilis* sporulation [65]. Through this strategy, φ29 harnesses the host's sensory systems to delay its own development under sub-optimal environmental conditions and to package its DNA into the endospore [65]. Indeed, there are data linking environmental regulatory systems of *C. crescentus* to its defense against φCbK infection via XRE TFs. Specifically, transposon insertions in the sensor histidine kinase, *skaH*, enhance host resistance to infection [30]. SkaH interacts with both the LovK and SpdS sensor histidine kinases, which in turn regulate SpdR [7]; SpdR activates expression of the XRE TFs studied here. Thus, variations in XRE TF expression may underlie the observed φCbK infection phenotype of *skaH* insertional mutants. The SpdS-SpdR system is believed to monitor shifts in the cellular redox state and electron transport chain flux [66–69] while LovK can function as both a photosensor and a redox sensor [70, 71]. Collective signaling from the SpdS-SkaH-LovK pathway, already recognized for its role in modulating *C. crescentus* cell adhesion [7], could also impact the interactions between *C. crescentus*, its environment, and its infecting bacteriophages.

## Materials and methods

### *Caulobacter* pangenome analysis

Nineteen *Caulobacter* genomes were analyzed in Anvi'o v7.1 [72] using the Snakemake pangenomics workflow [73]. Briefly, genome sequences NZ_CP073078, NZ_CP049199, NZ_CP048815, NZ_CP033875, NZ_CP026100, NZ_CP024201, NZ_CP013002, NZ_CP082923, NC_014100, CP096040, NC_01196, NC_002696, NC_010338, NZ_APMP01000001, NZ_CP023313, NZ_CP023314, NZ_CP023315, NZ_PEGF01000001, NZ_PEGH01000001 were retrieved from NCBI Genbank, reformatted using `anvi-script-reformat-fasta` and added to an anvi'o contigs database using `anvi-gen-contigs-database`. Open reading frames were identified using Prodigal [74], and predicted genes were functionally annotated with COG terms [75] and KEGG-KoFams terms [76], and annotation terms were added to the contigs database. Average nucleotide identity (ANI) was calculated using PyANI [77] with `anvi-compute-genome-similarity`. Gene clusters were identified using mcl [78] and the pangenome was generated using `anvi-pan-genome` with flags–mcl-inflation 3 and–min-occurrence 2. Pangenome summary data are presented in S1 Table.

### Bioinformatic analysis

Sequences for proteins from the *C. crescentus* NA1000 genome (GenBank accession number CP001340) that contained a HTH_XRE domain (cd00093) were extracted. Multiple sequence alignment was performed with Geneious Prime (version 2023.1.2) using Clustal Omega (version 1.2.2) with 1 iteration. Percent identity for pairwise alignments were calculated and plotted. For genomic neighborhood analysis, sequences were analyzed with the webFLaGs server (https://server.atkinson-lab.com/webflags) using the default parameters [79]. For S1 Fig, protein sequences were retrieved using the protein accession numbers and associated GCF assembly IDs for proteins from bins GC_0003, GC_0408, and GC_2778 in the pangenome analysis. For S2 Fig, protein sequences were retrieved using the protein accession numbers for a sampling of *cdxA* homologs modified from [25]) and the associated GCF assembly IDs. For S4 Fig, to identify CdxA homologs in phage, we performed a search with PSI-BLAST against the non-redundant protein sequence database (limited to Viruses—taxid:10239). Sequences of phage

proteins with e-value $< 1\times10^{-18}$ were extracted. CdxA homologs for hosts associated with the phage were extracted in the same manner. Sequences were aligned with Geneious Prime (version 2023.1.2) using Clustal Omega (version 1.2.2) with 100 iterations. A phylogenetic tree was constructed with the Geneious Tree Builder (genetic distance model: Jukes-Cantor; tree build model: Neighbor-joining; resampling method: bootstrap; number of iterations: 1000).

### Strain growth conditions

*Escherichia coli* was grown in Lysogeny broth (LB) or LB agar (1.5% w/v) at 37˚C [80]. Medium was supplemented with the following antibiotics when necessary: kanamycin 50 µg ml$^{-1}$, chloramphenicol 20 µg ml$^{-1}$, oxytetracycline 12 µg ml$^{-1}$, and carbenicillin 100 µg ml$^{-1}$.

*Caulobacter crescentus* was grown in peptone-yeast extract (PYE) broth (0.2% (w/v) peptone, 0.1% (w/v) yeast extract, 1 mM MgSO$_4$, 0.5 mM CaCl$_2$), PYE agar (1.5% w/v), or M2 defined medium supplemented with xylose (0.15% w/v) as the carbon source (M2X) [81] at 30˚C. Solid medium was supplemented with the following antibiotics where necessary: kanamycin 25 µg ml$^{-1}$, chloramphenicol 1 µg ml$^{-1}$, and oxytetracycline 2 µg ml$^{-1}$. Liquid medium was supplemented with the following antibiotics where necessary: chloramphenicol 1 µg ml$^{-1}$, and oxytetracycline 2 µg ml$^{-1}$.

### Plasmid and strain construction

Plasmids were cloned using standard molecular biology techniques and the primers listed in S5 Table. For overexpression constructs, inserts were cloned into pPTM057, which integrates at the xylose locus and contain a cumate-inducible (P$_{Q5}$) promoter [6]. For reporter constructs, inserts were cloned into pPTM056, which replicates in *C. crescentus*. Plasmids were transformed into *C. crescentus* by either electroporation or triparental mating [81]. Transformants generated by electroporation were selected on PYE agar supplemented with the appropriate antibiotic. Strains constructed by triparental mating were selected on PYE agar supplemented with the appropriate antibiotic and nalidixic acid to counterselect against *E. coli*. Gene deletions and allele replacements were constructed using a standard two-step recombination/counter-selection method, using *sacB* as the counterselection marker. Briefly, pNPTS138-derived plasmids were transformed into *C. crescentus* and primary integrants were selected on PYE/kanamycin plates. Primary integrants were incubated overnight in PYE broth without selection. Cultures were plated on PYE agar plates supplemented with 3% (w/v) sucrose to select for recombinants that had lost the plasmid. Mutants were confirmed by PCR amplification of the gene of interest from sucrose resistant, kanamycin sensitive clones.

### Analysis of transcription using fluorescent fusions

Strains were incubated in triplicate at 30˚C overnight in PYE broth supplemented with 1 µg ml$^{-1}$ chloramphenicol and 50 µM cumate. Overnight cultures were diluted in the appropriate broth supplemented with 1 µg ml$^{-1}$ chloramphenicol and 50 µM cumate to 0.01 OD$_{660}$ for the *cdxA* and *cdxB* reporters or 0.05 OD$_{660}$ for *gafYZ* reporters. Diluted cultures were incubated at 30˚C for 24 hours. For *hfiA* reporters, strains were inoculated in triplicate in M2X supplemented with 1 µg ml$^{-1}$ chloramphenicol and 50 µM cumate and grown overnight at 30˚C. Strains were subcultured and grown at 30˚C for 7 hours. Cultures were diluted to 0.0001– 0.005 OD$_{660}$ and incubated at 30˚C until reaching 0.05–0.1 OD$_{660}$. For measurements, 200 µl culture was transferred to a black Costar 96 well plate with clear bottom (Corning). Absorbance at 660 nm and fluorescence (excitation = 497 ± 10 nm; emission = 523 ± 10 nm) were measured in a Tecan Spark 20M plate reader. Fluorescence was normalized to absorbance. Statistical analysis was carried out in GraphPad 9.3.1.

## Holdfast imaging and quantification

Strains were inoculated in triplicate in M2X supplemented with 50 μM cumate and grown overnight at 30˚C. Strains were subcultured in M2X supplemented with 50 μM cumate and grown for 7 hours at 30˚C. Cultures were diluted to 0.0002–0.0053 $OD_{660}$ and incubated at 30˚C until reaching 0.05–0.1 $OD_{660}$. Alexa594-conjugated wheat germ agglutinin (WGA) (ThermoFisher) was added to the cultures with a final concentration of 2.5 μg ml$^{-1}$. Cultures were shaken at 30˚C for 10 min at 200 rpm. Then, 1.5 ml culture was centrifuged at 12,000 x g for 2 min, supernatant was removed, and pellets were resuspended in 35 μl M2X. Cells were spotted on 1% (w/v) agarose pads in $H_2O$ and imaged with a Leica DMI6000 B microscope. WGA staining was visualized with Leica TXR ET (No. 11504207, EX: 540–580, DC: 595, EM: 607–683) filter. Cells with and without holdfasts were enumerated using the image analysis suite, FIJI. Statistical analysis was carried out in GraphPad 9.3.1.

## Chromatin immunoprecipitation sequencing (ChIP-seq)

Strains were incubated in triplicate at 30˚C overnight in 10 ml PYE supplemented with 2 μg ml$^{-1}$ oxytetracycline when appropriate. Then, 5 ml overnight culture was diluted into 46 ml PYE supplemented with 2 μg ml$^{-1}$ oxytetracycline when appropriate and grown at 30˚C for 2 hours. Cumate was added to a final concentration of 50 μM and cultures were grown at 30˚C for 6 hours. Cultures were crosslinked with 1% (w/v) formaldehyde for 10 min, then crosslinking was quenched by addition of 125 mM glycine for 5 min. Cells were centrifuged at 7196 x g for 5 min at 4˚C, supernatant was removed, and pellets were washed in 25 ml 1x cold PBS pH 7.5 three times. Pellets were resuspended in 1 ml [10 mM Tris pH 8 at 4˚C, 1 mM EDTA, protease inhibitor tablet, 1 mg ml$^{-1}$ lysozyme] and incubated at 37˚C for 30 min. Sodium dodecyl sulfate (SDS) was added to a final concentration of 0.1% (w/v) and DNA was sheared to 300–500 bp fragments by sonication for 10 cycles (20 sec on/off). Debris was centrifuged at 15,000 x g for 10 min at 4˚C, supernatant was transferred, and Triton X-100 was added to a final concentration of 1% (v/v). Samples were pre-cleared through incubation with 30 μl SureBeads Protein A magnetic beads for 30 min at room temp. Supernatant was transferred and 5% lysate was removed for use as input DNA.

For pulldown, 100 μl Pierce anti-FLAG magnetic agarose beads (25% slurry) were equilibrated overnight at 4˚C in binding buffer [10 mM Tris pH 8 at 4˚C, 1 mM EDTA, 0.1% (w/v) SDS, 1% (v/v) Triton X-100] supplemented with 1% (w/v) bovine serum albumin (BSA). Pre-equilibrated beads were washed four times in binding buffer, then incubated with the remaining lysate for 3 hours at room temperature. Beads were washed with low-salt buffer [50 mM HEPES pH 7.5, 1% (v/v) Triton X-100, 150 mM NaCl], high-salt buffer [50 mM HEPES pH 7.5, 1% (v/v) Triton X-100, 500 mM NaCl], and LiCl buffer [10 mM Tris pH 8 at 4˚C, 1 mM EDTA, 1% (w/v) Triton X-100, 0.5% (v/v) IGEPAL CA-630, 150 mM LiCl]. To elute protein-DNA complexes, beads were incubated for 30 min at room temperature with 100 μl elution buffer [10 mM Tris pH 8 at 4˚C, 1 mM EDTA, 1% (w/v) SDS, 100 ng μl$^{-1}$ 3xFLAG peptide] twice. Elutions were supplemented with NaCl and RNase A to a final concentration of 300 mM and 100 μg ml$^{-1}$, respectively, and incubated at 37˚C for 30 min. Then, samples were supplemented with Proteinase K to a final concentration of 200 μg ml$^{-1}$ and incubate overnight at 65˚C to reverse crosslinks. Input and elutions were purified with the Zymo ChIP DNA Clean & Concentrator kit and libraries were prepared and sequenced at the Microbial Genome Sequencing Center (Pittsburgh, PA). Raw chromatin immunoprecipitation sequencing data are available in the NCBI GEO database under series accession GSE241057.

## ChIP-seq analysis

Paired-end reads were mapped to the *C. crescentus* NA1000 reference genome (GenBank accession number CP001340) with CLC Genomics Workbench 20 (Qiagen), ignoring non-specific matches. Peak calling was performed with the Genrich tool (https://github.com/jsh58/Genrich) on Galaxy; peaks are presented in S2 Table. Briefly, PCR duplicates were removed from mapped reads, replicates were pooled, input reads were used as the control dataset, and peak were called using the default peak calling option [Maximum q-value: 0.05, Minimum area under the curve (AUC): 20, Minimum peak length: 0, Maximum distance between significant sites: 100]. For subsequent analysis, ChIP-seq peaks with q-value $\leq 0.001$ were used. ChIPpeakAnno was used to determine the number of peaks that overlapped between the different XRE ChIP-seq datasets. Peaks were designated as 100 bp bins centered on the peak summit as identified by Genrich. Peaks were considered overlapping if at least 1 base pair overlapped.

## XRE motif discovery

For motif discovery, DNA sequences of ChIP-seq peaks from each dataset were submitted to the XSTREME module of MEME suite [82]. ChIP-seq peaks were designated as 100 bp bins centered on the summit coordinate identified by Genrich. For XSTREME parameters, the background model was constructed from shuffled input sequences (Markov Model Order: 2). The expected motif site distribution was set to 'zero or one occurrence per sequence'. Motifs length was designated as between 6 and 15 bp. The top MEME hit was designated as the enriched sequence motif.

## RNA preparation, sequencing, and analysis

Strains were incubated in triplicate at 30˚C overnight in 2 ml PYE supplemented with 50 μM cumate. Then, 1 ml overnight was diluted into 10 ml PYE supplemented with 50 μM cumate and grown at 30˚C for 4 hours (~0.18–0.26 $OD_{660}$). 8 ml culture was pelleted at 15,000 x g for 1 min at 4˚C, supernatant was removed, pellets were resuspended in 1 ml TRIzol, and stored at -80˚C. For φCbK infection time course RNA-seq, strains were incubated in triplicate at 30˚C overnight in 10 ml PYE. Then 2.5 ml overnight was diluted into 50 ml PYE and incubated at 30˚C for 4 hours (~0.18–0.2 $OD_{660}$), then diluted to 0.1 $OD_{660}$. For 0 min time point, 5 ml culture was pelleted at 15,000 x g for 1 min at 4˚C, supernatant was removed, pellets were resuspended in 1 ml TRIzol, and stored at -80˚C. Then, 3.9 x $10^{10}$ PFU φCbK (10 MOI) was added to cultures and cultures were incubated at 30˚C for 90 min while shaking. At 15, 30, 45, 60, 75, and 90 min, 5 ml culture was pelleted at 15,000 x g for 1 min at 4˚C, supernatant was removed, pellets were resuspended in 1 ml TRIzol, and stored at -80˚C. Samples were heated at 65˚C for 10 min and 200 μl chloroform was added. Samples were vortexed for 15 sec, then incubated at room temperature for 5 min. Samples were pelleted at 17,000 x g for 15 min at 4˚C, then the aqueous phase was transferred to a fresh tube. An equivalent volume 100% isopropanol was added to each sample, then mixed by inversion, and stored at -80˚C to precipitation nucleic acids. Thawed samples were then pelleted at 17,000 x g for 30 min at 4˚C and supernatant was removed. Samples were washed in 1 ml cold 70% ethanol, then vortexed briefly. Samples were then pelleted at 17,000 x g for 5 min at 4˚C, supernatant was removed, and pellets were air dried for 10 min. Pellets were resuspended in 100 μl RNase-free water and incubated at 60˚C for 10 min. Samples were treated with TURBO DNase and cleaned up with RNeasy Mini Kit (Qiagen).

Library preparation and sequencing was performed by SeqCenter with Illumina Stranded RNA library preparation and RiboZero Plus rRNA depletion (Pittsburgh, PA). Reads were

mapped to the *C. crescentus* NA1000 reference genome (GenBank accession CP001340) or φCbK reference genome (GenBank accession JX100813.1) using CLC Genomics Workbench (Qiagen). Differential gene expression was determined with CLC Genomics Workbench RNA-seq Analysis Tool. For φCbK *tgrL* overexpression, differential gene expression was designated as genes that significantly changed compared to the empty vector control (|fold-change| $\geq$ 1.5 and FDR p-value $\leq$ 0.0001). For φCbK infection time course, differential expression of *C. crescentus* genes was designated as genes that significantly changed in at least one time point compared to the 0 minute sample (uninfected) (|fold-change| $\geq$ 2 and FDR p-value $\leq$ 0.0001). For differential expression of φCbK genes during infection, fold-change was determined by comparison to the 15 minute sample. Raw sequencing data are available in the NCBI GEO database under series accession GSE241057.

To determine direct targets of host and phage XRE, ChIPpeakAnno was used to identify promoters that overlapped with XRE ChIP-seq peaks [83]. Promoters were designated as -300 to +100 bp around transcription start sites (TSS) (modified from [84]). For genes/operons that did not have an annotated TSS, the +1 residue of the first gene in the operon was designated as the TSS. Features were considered overlapping if at least 1 base pair overlapped. To cluster RNA-seq infection time course data, RPKM for each gene and time point was averaged and normalized by calculating the percent expression compared to maximum RPKM of the gene in the time course. Genes were then hierarchically clustered (clustering method: average linkage; similarity metric: uncentered correlation) using Cluster 3 [85] and viewed with Java Tree-View [86].

## Bacterial two-hybrid assay

The previously described bacterial two-hybrid system was utilized [87]. Plasmids containing fusions to T25 or T18c domains of adenylate cyclase were co-transformed into chemically competent BTH101 *E. coli* through heat shock at 42°C. Transformants were selected on LB agar supplemented with 50 μg ml$^{-1}$ kanamycin, 100 μg ml$^{-1}$ carbenicillin, 0.5 mM IPTG, and 40 μg ml$^{-1}$ X-gal. Strains were inoculated into 2 ml LB broth supplemented 30 μg ml$^{-1}$ kanamycin, 100 μg ml$^{-1}$ carbenicillin, and 0.5 mM IPTG, then shaken at 30°C overnight. Overnight cultures were diluted to 0.05 OD$_{600}$ and 5 μl diluted culture was spotted onto LB agar supplemented with 50 μg ml$^{-1}$ kanamycin, 100 μg ml$^{-1}$ carbenicillin, 0.5 mM IPTG, and 40 μg ml$^{-1}$ X-gal. Plates were imaged after growth at 30°C for 24 hours, followed by a 5 hour incubation at 4°C. For liquid culture assays, strains were inoculated into 2 ml LB broth supplemented 30 μg ml$^{-1}$ kanamycin, 100 μg ml$^{-1}$ carbenicillin, and 0.5 mM IPTG, then shaken overnight at 30°C. Then, 200 μl overnight culture was diluted into 2 ml LB broth supplemented 30 μg ml$^{-1}$ kanamycin, 100 μg ml$^{-1}$ carbenicillin, and 0.5 mM IPTG, grown at 30°C for 2 hours. OD$_{600}$ was measured and 100 μl culture was permeabilized with 100 μl chloroform. Then, 600 μl Z-buffer and 200 μl 4 mg ml$^{-1}$ ONPG was added to the reactions. Color development was stopped through the addition of 1 ml 1 M Na$_2$CO$_3$, OD$_{420}$ was measured, and Miller units were calculated.

## Phage infection assay

Strains were incubated at 30°C overnight in PYE supplemented with 50 μM cumate. Overnight cultures were diluted 1/10 in PYE supplemented with 50 μM cumate and incubated at 30°C for 4 hours. Cultures were diluted to 0.2 OD$_{660}$ in 1 ml PYE supplemented with 50 μM cumate and 1.3 x 10$^6$ φCbK phage were added (0.01 MOI). Phage were allowed to adsorb for 15 min at 30°C while shaking, diluted 1/1000 in PYE supplemented with 50 μM cumate, then shaken at 30°C for 180 min. For plating, 5 ml molten PYE top agar (0.3% (w/v) agar), 1 ml CB15 or

CB15 *xylX*::pPTM057 (0.2 OD$_{660}$), and 150 μl 30% (w/v) xylose were mixed and poured over a PYE plate. To enumerate the number of total phage (free + infected phage), 100 μl sample was plated on PYE top agar, then incubated at 30°C overnight. To determine the number of free phage, samples were treated with chloroform. Samples were diluted where appropriate, then 100 μl was plated on PYE top agar and incubated at 30°C overnight. Burst size was calculated as (Total PFU/ml $_{t = 180}$ –Free PFU/ml $_{t = 0}$)/(Total PFU/ml $_{t = 0}$ –Free PFU/ml $_{t = 0}$).

### Genomic DNA isolation

Strains were incubated at 30°C overnight in PYE supplemented with 50 μM cumate. Overnight cultures were diluted to 0.05 OD$_{660}$ in PYE supplemented with 50 μM cumate and incubated at 30°C for 24 hours. Cultures were pelleted at 12,000 x g for 1 min, supernatant was removed, pellets were washed with 0.5 ml H$_2$O, and then resuspended in 100 μl TE buffer (10 mM Tris pH 8.0, 1 mM EDTA). 500 μl GES lysis solution (5.08 M guanidium thiocyanate, 0.1 M EDTA, 0.5% (w/v) sarkosyl) was added, samples were vortexed, then heated at 60°C for 15 min. 250 μl 7.5 M cold ammonium acetate was added, samples were vortexed for 15 sec, and incubated on ice for 10 min. 500 μl chloroform was added and samples were vortexed for 15 sec. Samples were pelleted at 12,000 x g for 10 min, then the aqueous phase was transferred to a fresh Eppendorf tube. 324 μl cold isopropanol (0.54 volumes) was added, samples were mixed by inversion, and incubated at room temperature for 15 min. Samples were pelleted at 12,000 x g for 3 min, supernatant was removed, and pellets were washed with 700 μl 70% (v/v) ethanol. Pellets were air dried for 10 min, then resuspended in 100 μl TE buffer. DNA concentrations were measured by NanoDrop. Samples were diluted to 400 ng/μl and 20 μl diluted sample was run on a 1% (w/v) agarose gel. Gels were imaged on a ChemiDoc MP with 605/50 filter and UV trans illumination.

### Supporting information

**S1 Fig. XRE transcription factor paralogs have conserved genomic neighborhoods in the Caulobacter species.** Genomic neighborhood analysis of XRE transcription factor paralogs. Phylogenetic tree based on XRE transcription factor sequences (left) and genomic neighborhood surrounding those genes (right). Protein sequences were retrieved using the protein accession numbers and associated GCF assembly IDs for proteins from bins GC_0003, GC_0408, and GC_2778 in the pangenome analysis (Fig 2A) and analyzed with the webFLaGs server (https://server.atkinson-lab.com/webflags) [79]. Numbers on the phylogenetic tree indicate bootstrap values. XRE homologs are colored black, orthologous genes are colored and numbered identically, non-conserved genes are uncolored and outlined in grey, pseudogenes are uncolored and outlined in blue, and non-coding RNA genes are uncolored and outlined in green.
(TIF)

**S2 Fig. *cdxA* homologs and surrounding genomics neighborhood are conserved across Alphaproteobacteria.** Phylogenetic tree based on XRE transcription factor sequences (left) and genomic neighborhood surrounding those genes (right). Protein accession numbers (were modified from [25]) and associated GCF assembly IDs were analyzed with the webFLaGs server (https://server.atkinson-lab.com/webflags) [79]. Numbers on the phylogenetic tree indicate bootstrap values. *cdxA* homologs are colored black, orthologous genes are colored and numbered identically, non-conserved genes are uncolored and outlined in grey, pseudogenes are uncolored and outlined in blue, and non-coding RNA genes are uncolored and outlined in green.
(TIF)

**S3 Fig. RtrC activates expression of *cdxA* and *cdxB*. A & C)** RtrC binds the *cdxA* and *cdxB* promoter *in vivo*. ChIP-seq profile from pulldowns of 3xFLAG-tagged protein are shown. Lines indicate the fold-enrichment from pulldowns compared to an input control. Genomic position and relative position of genes are indicated. Data are in 25 bp bins and are the mean of three biological replicates. **B & D)** *cdxA* and *cdxB* expression using a $P_{cdxA}$- or $P_{cdxB}$-*mNeon-Green* reporter. Fluorescence was measured in either a wild type background containing either an empty vector (EV) or *rtrC* overexpression (++) vector. Fluorescence was normalized to cell density. Data are the mean and error bars are the standard deviation of three biological replicates. Statistical significance was determined by multiple unpaired t-test using the Holm-Šídák method to correct for multiple comparisons (p-value ≤ 0.0001,****).
(TIF)

**S4 Fig. XRE homologs are encoded by Alphaproteobacterial phage.** Phylogenetic tree (left) and multiple sequence alignment (right) of XRE homologs from different *Alphaproteobacteria* and Alphaproteobacterial phage. Numbers above branches indicate percent bootstrap support and branch length corresponds to substitutions per site. Protein accession numbers and organism are displayed next to corresponding branches. Alignments (right) match the order in the phylogenetic tree (left). For alignments, horizontal lines indicate gaps. Pink rectangle indicates the location of the HTH_XRE domain.
(TIF)

**S5 Fig. φCbK genes are expressed in distinct temporal patterns throughout infection. A)** Hierarchical clustering of φCbK gene expression during infection of *C. crescentus*. Relative values (i.e. % max expression) were calculated by normalizing transcript levels at a time point to the maximum transcript levels for that gene over the infection time course. Relative gene expression was hierarchically clustered using Cluster 3.0 [85] and plotted as a heatmap. Rows correspond to φCbK genes and clusters are colored and labeled. Data are the mean of three biological replicates. **B-G)** Relative gene expression of clusters from hierarchical clustering. Data are the mean relative expression of all genes within the indicated cluster and error bars are the associated standard deviations. Wild type cells were infected during logarithmic growth phase in complex medium (PYE) at 10 multiplicity of infection (MOI).
(TIF)

**S6 Fig. *C. crescentus* and φCbK XRE proteins form homo- and heteromeric interactions.** Interaction between RtrA, RtrB, CdxA, CdxB, StaR, and φCbK TgrL based on bacterial two-hybrid (BTH) assays. Proteins were fused to split adenylate cyclase fragments (T18c and T25) and co-expressed. Interactions between the fused proteins reconstitutes adenylate cyclase, promoting expression of a *lacZ* reporter. Empty vector (EV) are the negative control and Zip is the positive control. β-galactosidase activity was measured, and Miller units were calculated. Data are the mean and error bars are the standard deviation of at least three biological replicates. Statistical significance was determined by one-way ANOVA compared to the EV only control followed by Dunnett's multiple comparison. Non-significant columns are indicated with (ns). Columns with p-value ≤ 0.05 were indicated with (*). All other columns had p-values ≤ 0.0001.
(TIF)

**S1 Table. Pangenome analysis results.**
(XLSX)

**S2 Table. ChIP-seq data.**
(XLSX)

**S3 Table. ΦCbk infection RNA-seq data.**
(XLSX)

**S4 Table. RNA-seq and ChIP-seq analysis.**
(XLSX)

**S5 Table. Strains and Plasmids.**
(XLSX)

# Acknowledgments

We thank members of the Crosson lab, as well as Tung Le and Emma Banks for valuable feedback and discussions throughout the course of this study.

# Author Contributions

**Conceptualization:** Maeve McLaughlin, Aretha Fiebig, Sean Crosson.

**Formal analysis:** Maeve McLaughlin.

**Funding acquisition:** Maeve McLaughlin, Sean Crosson.

**Investigation:** Maeve McLaughlin.

**Methodology:** Maeve McLaughlin.

**Validation:** Maeve McLaughlin.

**Visualization:** Maeve McLaughlin, Aretha Fiebig, Sean Crosson.

**Writing – original draft:** Maeve McLaughlin, Sean Crosson.

**Writing – review & editing:** Maeve McLaughlin, Aretha Fiebig, Sean Crosson.

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
