## [Decision Letter · Decision Letter 0]

20 Sep 2023

Dear Sean,

Thank you very much for submitting your Research Article entitled 'XRE Transcription Factors Conserved in Caulobacter and φCbK Modulate Adhesin Development and Phage Production' to PLOS Genetics.

The manuscript was fully evaluated at the editorial level and by independent peer reviewers. As you will see, both reviewers are enthusiastic; there are some minor concerns that we ask you address in a revised manuscript.

We therefore ask you to modify the manuscript according to the review recommendations. Your revisions should address the specific points made by each reviewer.

Yours sincerely,

Gregory S. Barsh

Editor-in-Chief

PLOS Genetics

Gregory Copenhaver

Editor-in-Chief

PLOS Genetics

Reviewer's Responses to Questions

**Comments to the Authors:**

Reviewer #1: Summary:

This manuscript from McLaughlin et al. reports on the family of XRE type-transcription factors in Caulobacter crescentus and in related phages. This family has undergone an astounding expansion in different lineages of Caulobacter and in related Alphaproteobacteria. The study does a very nice genomic comparison and then drills down into the functions for a few of these in C. crescentus, focusing most on the PhiCbK phage encoded regulator. The overlapping specificity and pervasiveness of the XRE family is intriguing, and does invoke interesting ideas such as the concept of ecoparalogs and regulatory feedbacks in bacterial and phage systems.

Major Comments:

This is an excellent study. The experimentation and comparative bioinformatics is compelling and well performed, and the writing is clear and insightful. I have no major qualms with the quality of the data, or the information reported.

I would suggest that the authors spend a bit more time explaining and perhaps diagramming the structure of the XRE transcription factors. They are treated as regulatory actors rather than as complex molecules in the paper, and a bit more info on how they function would strengthen the manuscript. They are short proteins comprised predominantly of a HTH motif. CI and Cro have been beaten to death and so by comparison, the other XREs may work similarly? A little more info on projected mechanism of action would be helpful.

Minor Comments:

1. L124. It would be worth mentioning that holdfast production in M2X is usually quite low due to the inhibitory activity of hfiA among other things, compared with rich medium such as PYE. Otherwise, readers may question the strikingly low low values in Fig. 4B.

2. L244-45. Were these all the same positions for the fusions? N-terminal or C-terminal with T18/T25.

Reviewer #2: This manuscript by McLaughlin, et al explores members of the XRE family of transcription factors identified through a pangenome analysis of Caulobacter. Using ChIP-seq and RNA seq, they show that XRE proteins encoded in Caulobacter and in the phiCbK phage can control transcription of overlapping sets of genes including holdfast regulators and gene transfer agents. Additional RNAseq experiments support roles for the phage encoded XRE being expressed early stages of infection, suggesting an important role for this factor in phage infectivity. These XRE proteins seem to be quite redundant and ectopic expression of any of the paralogs in a ∆xre background can increase phage burst size and suppress gafYZ/hfiA to varying degrees. The XRE proteins interact with themselves and each other based on bacterial two-hybrid results and the authors discuss how this heteromeric interaction may lead to complex regulation. The authors conclude that XRE genes play multiple roles, including adhesion regulation and host-phage dynamics during infection. Overall, the work is extensive, resulting in a wealth of new binding site data and transcriptional results, the experiments are well done, and the interpretations are clearly supported by the data. Some lingering questions remain as described below, the first three of which in this reviewers' opinion could benefit from experiments (but not critically), while the fourth could be addressed relatively easily.

1. It was ultimately unclear what the link was between XRE function in the host (holdfast production, GTA regulation, etc) and why the phage encoded TgrL would be important to express early for the infection. The authors addressed this discordance in the discussion, where they mention that phage infection does not lead to change in expression of the holdfast regulators (line 328 and table S3). It is appreciated that this would be challenging to fully address, but the absence of experimental connection between the molecular pathways controlled by XREs and the role of TgrL in phage infections slightly reduces the overall enthusiasm. Of course, if the authors show that ∆tgrL phage are not deficient in infectivity, then that would also clarify this point in that it would show this gene is not important for that process.

2. Similarly, what role do host XREs play in the phage burst size effects? Based on the RNAseq, the XREs themselves are not induced upon phage infection either, so why would deletion of these factors reduce burst size by 50% (Figure 9). Could this be a more indirect effect? For example, are there changes in growth, development, or cell surface that make ∆xre strains less susceptible? Is there a change in the timing of lysis either seen in population measurements or microscopically? Again, it is appreciated that this may be challenging to address experimentally, but at a minimum, a deeper description of this aspect of the host XREs in phage virulence would be good in the discussion.

3. They authors mention that cdxB was a hit in the Christen study looking at genes important for phiCbK infection. Were any of the other XREs also implicated? If not, is cdxB the major driver of phenotypes for the ∆xre strain? This should be brought up in the text or tested.

4. The authors make a compelling argument that XRE proteins can form heteromers based on their bacterial two hybrid results. As described in the discussion, this additional multimerization could lead to a rich control of regulation by different combinations of XREs. However, almost because it was referred to so much, it would be useful to have some experimental data to back up this claim. For example, the authors note that TgrL XRE interacts with itself as well as it does with CdxA and CdxB. Does ectopic expression of TgrL in the presence of either of these XREs change the regulation of hfiA or gafXY? This could be addressed with either co-expression experiments, or in genetic backgrounds like the ∆xre strain and adding back one or both of the CdxA/B encoding genes. This could be done for other combinations of XRE as well (or instead of the TgrL work) but some experimental evidence for the importance of XRE heteromeric activity would strengthen the impact of this aspect of the work substantially.

5. The authors are commended for laying out a wonderfully clear depiction of ideas and results. The supplemental tables are extensive but allow afficionados to see the details without losing the importance of the bigger picture. This attention to all aspects of this work are appreciated.

minor:

line 168 should be Figure 6C I think.

**Have all data underlying the figures and results presented in the manuscript been provided?**

Reviewer #1: Yes

Reviewer #2: Yes

PLOS authors have the option to publish the peer review history of their article (what does this mean?). If published, this will include your full peer review and any attached files.

Reviewer #1: **Yes: **Clay Fuqua

Reviewer #2: No

---

## [Editor Report · Decision Letter 1]

3 Nov 2023

Dear Sean,

We are pleased to inform you that your manuscript entitled "XRE Transcription Factors Conserved in Caulobacter and φCbK Modulate Adhesin Development and Phage Production" has been editorially accepted for publication in PLOS Genetics. Congratulations!

Yours sincerely,

Gregory S. Barsh

Editor-in-Chief

PLOS Genetics

Gregory Copenhaver

Editor-in-Chief

PLOS Genetics

Comments from the reviewers (if applicable):

**Data Deposition**

http://datadryad.org/submit?journalID=pgenetics&manu=PGENETICS-D-23-00944R1

**Press Queries**

---

## [Editor Report · Acceptance letter]

10 Nov 2023

PGENETICS-D-23-00944R1 

XRE Transcription Factors Conserved in Caulobacter and φCbK Modulate Adhesin Development and Phage Production 

Dear Dr Crosson, 

We are pleased to inform you that your manuscript entitled "XRE Transcription Factors Conserved in Caulobacter and φCbK Modulate Adhesin Development and Phage Production" has been formally accepted for publication in PLOS Genetics! Your manuscript is now with our production department and you will be notified of the publication date in due course.

With kind regards,

Anita Estes

PLOS Genetics

On behalf of:
